# Expanding tracer space for positron emission tomography with high molar activity $^{18}$F-labeled α,α-difluoromethylalkanes

Qunchao Zhao, Sanjay Telu ✉, Shuiyu Lu & Victor W. Pike ✉

Positron emission tomography (PET) is an advanced biomedical imaging modality that relies on well-designed radiotracers to report on specific protein targets and processes occurring in living animals and humans. Cyclotron-produced short-lived fluorine-18 ($t_{1/2} = 109.8$ min) is widely used to radiolabel tracers for PET. Herein we aim to expand the chemical space available for PET tracer development to include structures with $^{18}$F-labeled α,α-difluoromethylalkyl groups. We report an efficient and broad-scope method for labeling such groups with high molar activities based on a single-step radiofluorination of α-bromo-α-fluoroalkanes. The method is applicable to bioactive compounds and drug-like molecules, and is readily automated for radiotracer production. The unique physical and biochemical features of the α,α-difluoromethyl group can now be exploited in the design of new PET tracers.

Positron emission tomography (PET) is an advanced imaging modality that is widely applied in biomedical research, drug development, and clinical diagnosis. The broad utility of PET derives from well-designed radiotracers that can report on specific protein targets and processes occurring in animals and humans in vivo[1,2]. Among positron-emitting radionuclides, fluorine-18 ($^{18}$F) is one of the most commonly used for PET[3]. This is not only because of its high accessibility in a no-carrier-added (NCA) state from modern cyclotrons through the high-yielding $^{18}$O(p,n)$^{18}$F reaction on [$^{18}$O]water[4] but also because of its many favorable physical properties for imaging and tracer design. These properties include: 1) almost pure decay ($\beta^+$, 97%) by emission of a relatively low energy positron ($\beta^+$, 0.634 MeV) enabling acquisition of images with high spatial resolution (ca 2 mm)[5]; 2) a usefully long half-life ($t_{1/2} = 109.8$ min) that i) allows time for efficient incorporation into tracers and their distribution to nearby standalone PET imaging facilities, and ii) offers the possibility to monitor tracer kinetics in vivo over a few hours[6]; 3) a van der Waals radius similar to that of hydrogen that leads to only a minor steric perturbation when fluorine-18 replaces hydrogen

at carbon; and 4) high electronegativity, enabling the alteration of molecular electrostatics and overall polarity.

Indeed, since the late 1970s, fluorine-18 has featured prominently in a vast array of successful PET tracers[7,8]. The most frequently used methods for their preparation are based on aliphatic and aromatic nucleophilic substitution reactions on substrates with good leaving groups[3,9]. These methods are well-established for their abilities to generate $^{18}$F-labeled tracers in acceptable yields and with high molar activities (radioactivity per mole of all isotopologues, denoted $A_m$)[10]. However, the success achieved in constructing C($sp^2$)–$^{18}$F and C($sp^3$)–$^{18}$F bonds has not been readily translated to $^{18}$F-labeled compounds containing multiple fluorine atoms, especially when they are situated on the same carbon atom, such as an α,α-difluoromethyl moiety (RCF$_2$H).

Interest in the development of methods for incorporating the α,α-difluoromethyl group (−CF$_2$H) into modern pharmaceuticals and agrochemicals has surged. Examples of such bioactive molecules appear in Fig. 1A. This interest is driven by a recognition of the

Molecular Imaging Branch, National Institute of Mental Health, National Institutes of Health, 10 Center Drive, Bethesda, MD 20892–1003, USA.
✉e-mail: sanjay.telu@nih.gov; pikev@mail.nih.gov

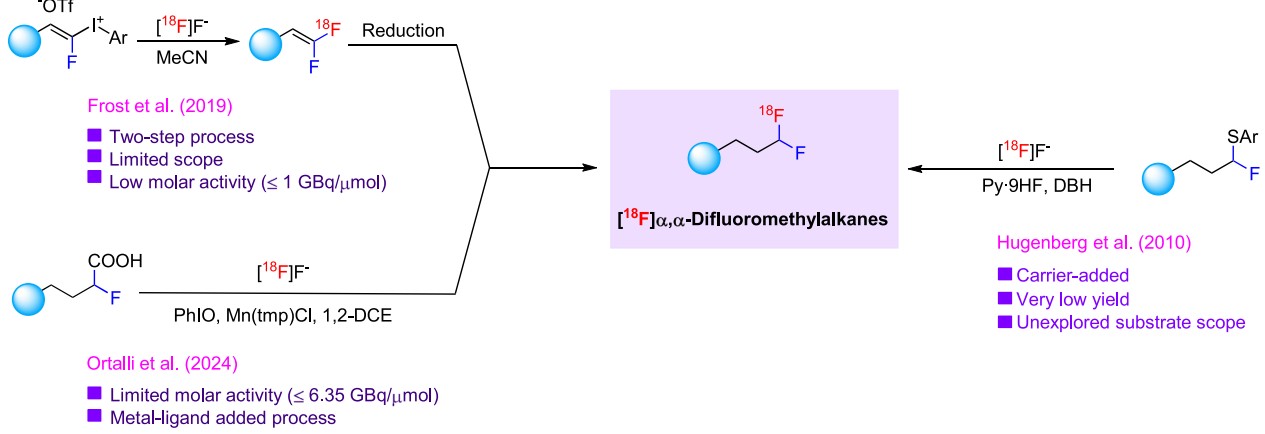

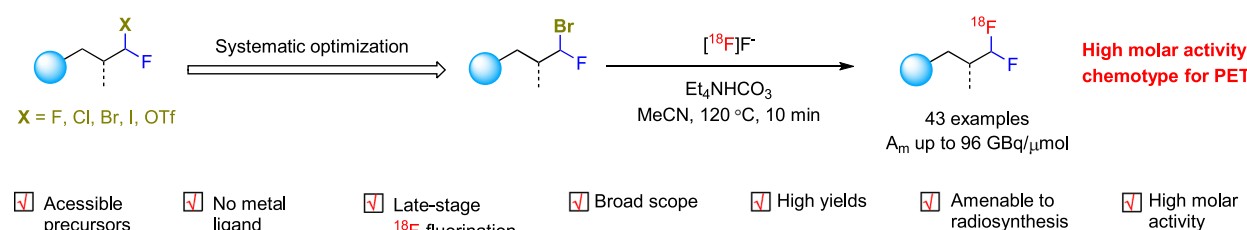

**Fig. 1 | Biomolecules containing α, α-difluoromethyl groups and approaches for their labeling with fluorine-18. A** Examples of biomolecules containing α,α-difluoromethylalkyl groups. **B** Prior methods for accessing [18]F-labeled α,α-

difluoromethylalkyl groups and their limitations. **C** Methodology for synthesizing [18]F-labeled α,α-difluoromethylalkyl groups at high molar activity, as presented in this work. $A_m$ molar activity, PET positron emission tomography.

potential of the difluoromethyl group to confer distinct advantages in biomolecule designs[11,12]. For example, the highly polarized carbon-hydrogen bond in the difluoromethyl group endows this group with the capability to act as a hydrogen bond donor, an attribute that sets it apart from other polyfluorinated chemical motifs[13,14]. Moreover, difluoromethyl groups can serve as metabolically stable bioisosteres of hydroxy, thiol, or amino groups and can therefore be used to improve pharmacokinetic and physicochemical properties[15,16].

The demand for novel methods to access the [18]F-labeled α,α-difluoromethyl moiety and thereby to expand the chemical space for PET tracer development also continues to grow[11]. A formidable challenge in this area is to meet the need for a straightforward general method that is capable of delivering good tracer yields along with usefully high molar activities. Usually, methods give low molar activity because a pre-existing C−F bond in the precursor for radiofluorination is vulnerable to cleavage during the labeling process to produce appreciable amounts of carrier fluoride ion as a competing reagent in the reaction. In the past decade, notable progress has been made in synthesizing [18]F]difluoromethylated arenes and heteroarenes from

[18]F]fluoride[17–22]. However, these methods variously require specialized precursors, metal catalysts, or additional synthesis manipulations after radiofluorination, and they all provide mediocre molar activities for the [18]F-difluoromethylarene products. Other methods, based on generating the [18]F]difluoromethyl radical[23] or [18]F]difluorocarbene[24], have enabled [18]F-difluoromethylarenes to be obtained with improved molar activity. Although methods to access [18]F]difluoromethylarenes are therefore well-established[25–27], they are nonetheless unsuitable for the efficient syntheses of [18]F-labeled α,α-difluoromethylalkanes.

After 2020, methods for attaching non-radioactive difluoromethyl groups to aliphatic chains were achieved with various difluoromethylating agents, such as the zinc complex, $(DMPU)_2Zn(CF_2H)_2$, difluoroacetic acid, chlorodifluoromethane, ((difluoromethyl)sulfonyl) arenes, difluoromethyltriphenylphosphonium iodide, and N-phenyl-N-tosyldifluoroacetamide[28–37]. Nevertheless, these methods are not adaptable to fluorine-18 labeling due to their need for harsh reaction conditions, long reaction times, and likely outcomes of high carrier dilution. To date, few methods exist for the radiosynthesis of an [18]F-labeled α,α-difluoromethylalkane. The earliest report showed only one example

**A**

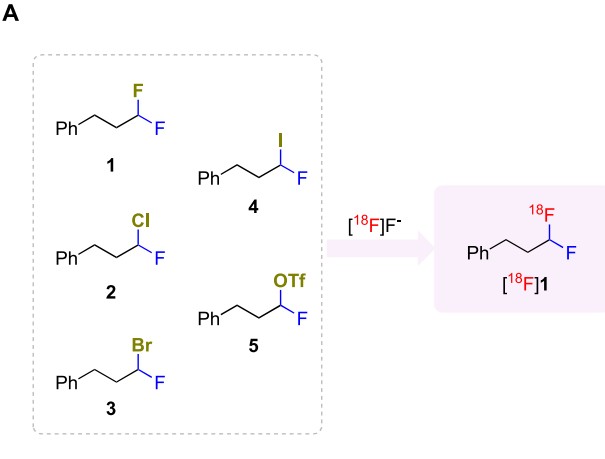

**B**

| Entry | X | Yield of **1** (%) |
|---|---|---|
| 1 | F | N/A |
| 2 | Cl | ND |
| 3 | Br | ND |
| 4 | I | ND |
| 5 | OTf | 11 |
| 6[a] | OTf | 10 |

[a]Performed at room temperature.

**C**

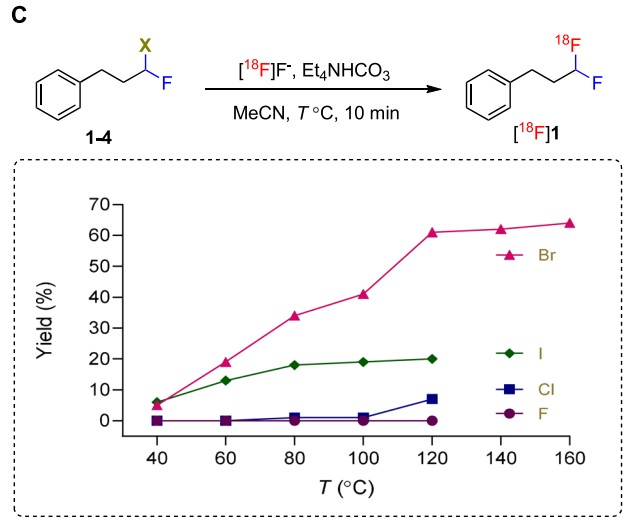

**D**

| Entry | Conditions | Yield of [$^{18}$F]**1** (%) |
|---|---|---|
| 1 | Starting[a] | 61 ± 8 (n=2) |
| 2 | $K_2CO_3$/K 2.2.2 instead of $Et_4NHCO_3$ | 47 ± 1 (n=2) |
| 3 | DMF instead of MeCN | 6 ± 0 (n=2) |
| 4 | DMSO instead of MeCN | 2 ± 1 (n=2) |
| 5 | 1.0 mg of $Et_4NHCO_3$ was used | 37 ± 6 (n=2) |
| 6 | 5 μmol precursor was used | 66 ± 8 (n=2) |
| 7 | 2.5 μmol precursor was used | 67 ± 5 (n=3) |

[a]As described in legend.

**Fig. 2 | Optimization study for the radiosynthesis of [$^{18}$F]1. A** Precursors designed for the radiosynthesis of [$^{18}$F]**1**. **B** Study of fluoride leaching from precursors **1**–**5** under typical radiofluorination conditions but in the absence of added fluoride. Yields were determined by $^{19}$F-NMR analysis of the crude reaction mixtures. N/A not applicable, ND not detected. **C** Radiofluorination of α-halo precursors **1**–**4** at different temperatures. Decay-corrected yields are calculated from HPLC analyses of crude reaction mixtures; no other $^{18}$F-labeled fluoro products were detected other than [$^{18}$F]**1**. **D** Optimization of the synthesis of [$^{18}$F]**1** from 3-bromo precursor **3**. Starting conditions: a solution of [$^{18}$F]fluoride (50–100 MBq) and tetra-ethylammonium bicarbonate (0.5 mg) in acetonitrile (200 μL) was added to a 1-mL vial followed by a solution of precursor (10 μmol) in acetonitrile (300 μL) and heated at 120 °C for 10 min. Yields are mean ± SD for n ≥ 2.

with a low yield from an α-fluoroalkyl aryl thioether precursor by the combined action of the oxidant, 1,3-dibromo-5,5-dimethylhydantoin, and a large excess of carrier-added pyridine·9H[$^{18}$F]F[38] (Fig. 1B). More recently two methods have appeared[39,40]. One method requires two steps from an aryldifluorovinyl iodonium salt and gives a low molar activity[39]. Another method is single-step from a 2-fluoroalkanoic acid precursor but gives only somewhat improved molar activity[40]. This method also requires the use of an oxidizing agent and a metal-ligand mediator. Thus, there is a clear need for the development of simple and efficient broad-scope methodologies for labeling α,α-difluoromethylalkanes with fluorine-18 at high molar activity.

We envisioned that [$^{18}$F]fluoride treatment of an α-fluoroalkane possessing a leaving group on the α-carbon might provide an $^{18}$F-labeled α,α-difluoromethylalkane with high molar activity. Herein, we describe our development of an effective method for the direct radiofluorination of α-bromo-α-fluoroalkane precursors with [$^{18}$F] fluoride, enabling the efficient late-stage syntheses of $^{18}$F-labeled α,α-difluoromethylalkanes with high molar activities. This method has several other attractive features, including amenability for automation

(Fig. 1C). We further demonstrate robust applications of this methodology for installing an $^{18}$F-labeled α,α-difluoromethyl moiety into diverse candidate PET tracers.

## Results and discussion
### Optimization study for late-stage radiofluorination
Nucleophilic substitutions are the most straightforward methods to be considered for labeling compounds on $sp^3$ carbon with cyclotron-produced [$^{18}$F]fluoride ion. However, the preparation of $^{18}$F-labeled α,α-difluoromethylalkanes by nucleophilic radiofluorination has not been explored previously. Therefore, for trial radiofluorinations, we first opted to design and screen an array of model precursors bearing both a fluorine atom and a leaving group in α-position on a phenyl-propyl skeleton (Fig. 2A). However, we expected that a few challenges would emerge, namely: 1) the presence of the α-fluorine atom might risk release of carrier fluoride during the radiofluorination, potentially leading to a dilution of molar activity[18–21]; 2) the α-fluorine atom might impede nucleophilic substitution[41]; and 3) the α-fluoro substituent might enhance a competing elimination pathway[41]. We sought to

address these challenges via systematic optimization of the radio-fluorination conditions with the model precursors, 1–5 (Fig. 2B–D).

Achieving high molar activity is essential for PET tracers to ensure compliance with the tracer principle and to minimize undesirable carrier occupancy at low density imaging targets by excessive carrier[10,42]. Therefore, we began our investigation by using [19]F-NMR spectroscopy to check the stability of the precursors 1–5 under basic conditions commonly used for radiofluorination, except we did not add a source of fluoride.

The C–F bond in (3,3-difluoropropyl)benzene (1) did not cleave or form any fluorinated elimination products when heated in acetonitrile at 120 °C, demonstrating that this α,α-difluoromethylalkane is stable under these conditions (entry 1, Fig. 2B). We also monitored the possible formation of 1 from the reactions of 2–5 with any fluoride ion that might be leached from the precursor in the presence of tetraethylammonium bicarbonate (Fig. 2B). Gratifyingly, under these conditions, (3-chloro-3-fluoropropyl)benzene (2), (3-bromo-3-fluoropropyl)benzene (3), and (3-fluoro-3-iodopropyl)benzene (4) did not produce any detectable difluoromethyl compound 1 (entries 2–4, Fig. 2B). The halo precursors 3 (X = Br) and 4 (X = I) however did appear to produce some fluorine-containing elimination product (as an E/Z isomer mixture), as evidenced by [19]F-NMR resonances at –129.6 and –131.6 ppm (Supplementary Figs. 3 and 4). 1-Fluoroalkyl triflates are known to have high reactivity towards nucleophiles[43]. Treatment of 1-fluoro-3-phenylpropyl triflate (5) at 120 °C or room temperature gave 1 in about 10% yield (entries 5 and 6, Fig. 2B), implying that some release of fluoride occurs. The radiofluorination of an α-fluoro-α-triflyl precursor, such as 5, would therefore be expected to give a radiolabeled product with low molar activity. Our finding is consistent with a report that the treatment of 1-fluorononyl triflate with the base, 1,8-diazabicyclo[5.4.0]undec-7-ene (DBU), slowly produces α,α-difluorononane because DBU extracts fluoride ion from the substrate by an $S_N2$ reaction; this fluoride then rapidly displaces triflate from another molecule of 1-fluorononyl triflate, as depicted in Supplementary Fig. 7[43]. We therefore focused our further investigations on the halo precursors, 2–4.

We next explored reaction stoichiometry using 3 as a model substrate. Reactions of 3 (0.05 mmol) in acetonitrile at 120 °C for 10 min, with the number of equivalents of tetramethylammonium fluoride varied between zero and one, were monitored by [19]F-NMR using (1,1,1-trifluoromethyl)benzene as an internal standard. With no externally added fluoride, only the unreacted precursor 3 was detected by [19]F-NMR (δ –131.1 ppm) (Supplementary Fig. 8). Otherwise, irrespective of the number of fluoride equivalents, the product consisted of about 10% of the desired α,α-difluoromethylalkane 1 (δ –116.4 ppm) and 25–30% of the elimination products E/Z-1b (δ –129.58 and –131.60 ppm) (Supplementary Table 1 and Supplementary Figs. 9–11). These results indicated that substitution of a higher halo substituent was likely a viable route for producing NCA [18]F-labeled α,α-difluoromethylalkanes from [18]F]fluoride, where the precursor would be in very large molar excess over [18]F]fluoride, but not a practical route for producing non-radioactive α,α-difluoromethylalkanes in high yields.

The feasibility of the radiofluorination of the halo precursors was first investigated with [18]F]fluoride as the limiting reagent in acetonitrile using tetraethylammonium bicarbonate as the base between 40 and 120 °C (Fig. 2C). As a control reaction, difluoro precursor 1 did not give any [18]F]1 under all examined temperatures, further evidencing its strong resistance to [19]F for [18]F exchange. The chloro precursor 2 provided only a low yield of [18]F]1, even at 120 °C. The yield of [18]F]1 increased to 18% with iodo precursor 4 at 80 °C. However, there was no notable increase in yields at higher temperatures. Interestingly, the bromo precursor 3 displayed remarkable performance. Radiofluorination yields of [18]F]1 steadily increased as the reaction temperature was raised. [18]F]1 could be obtained in 61% yield within 10 minutes at 120 °C.

The greater reactivity of the bromo precursor 3 over that of the iodo precursor 4 deserves comment. The leaving group abilities of halo substituents in aliphatic nucleophilic substitution reactions normally follow the order I > Br > Cl > F. The observed higher performance of the bromo precursor (3) over the iodo precursor (4) could be because the electron cloud of the iodo substituent is more polarizable than that of the bromo substituent. As a result, the electron cloud on iodine may be more easily drawn towards the carbon by the electronegative α-fluorine substituent, thereby increasing the electron density on the carbon center and making it less susceptible to nucleophilic attack by [18]F]fluoride ion.

For further radiofluorination optimization, we focused on the high-performing bromo precursor 3. We explored the effect of higher temperature. However, the yield of [18]F]1 did not increase substantially between 120 °C and 160 °C (Fig. 2C and Supplementary Table 2). Thus, further optimizations, looking at base, solvent, and amount of precursor, were performed at 120 °C to keep the reaction conditions relatively mild, especially when using low boiling-point acetonitrile as solvent (Fig. 2D).

The use of potassium carbonate-kryptofix 2.2.2 (K 2. 2. 2) as base, as is used widely in radiofluorination[3] instead of tetraethylammonium bicarbonate, gave [18]F]1 in 47% yield (entry 2, Fig. 2D). Switching the reaction solvent from acetonitrile to an alternative polar aprotic solvent, such as N,N-dimethylformamide or dimethylsulfoxide, was found to be detrimental (entries 3 and 4, Fig. 2D). Doubling the amount of tetraethylammonium bicarbonate significantly lowered the yield of [18]F]1 (entry 5, Fig. 2D). In this case, excess base may have increased decomposition of the precursor by elimination at the expense of radiofluorination. Remarkably, reducing the amount of precursor 3 from 10 μmol to a low loading of 5 μmol and then 2.5 μmol did not curtail yield; [18]F]1 was formed in 66% and 67% yields, respectively (entries 6 and 7, Fig. 2D). The conditions in entry 7 of Fig. 2D (2.5 μmol precursor, 0.5 mg Et₄NHCO₃, 500 μL MeCN, 120 °C, 10 min) were therefore considered optimal.

## Substrate scope for the late-stage [18]F-fluorination of α-bromo-α-fluoroalkanes

Having optimized the conditions for synthesizing the model compound [18]F]1 (entry 7, Fig. 2D), we next investigated the substrate scope for this late-stage radiofluorination methodology. Precursor, α-bromo-α-fluoroalkanes of wide structural diversity were synthesized either by late-stage one-step photoredox-catalyzed addition of dibromofluoromethane to a terminal alkene[44] or by multi-step synthesis starting from an appropriate aldehyde (Supplementary Information, Section 2.4.). The latter involved the treatment of the aldehyde with triflic anhydride followed by sequential nucleophilic fluorination and bromination[45]. The reference α,α-difluoromethylalkanes were synthesized by similar methods (Supplementary Information, Section 2.4).

To our delight, a diverse array of the precursors reacted with [18]F]fluoride to give the desired [18]F-labeled α,α-difluoromethylalkanes in moderate to excellent yields (Fig. 3). Importantly, the electronic properties of aryl substituents did not significantly influence yields. Precursors featuring a para electron-donating aryl substituent, such as tert-butyl ([18]F]6), methoxy ([18]F]7, [18]F]17), or phenyl ([18]F]10), worked well, furnishing [18]F-labeled α,α-difluoromethyl compounds in over 50% yields. Meta- or para- electron-withdrawing substituents, including methylsulfonyl ([18]F]12), nitro ([18]F]13), nitrile ([18]F]14, [18]F]20), ester ([18]F]15), and trifluoromethyl ([18]F]21), were well tolerated. Moreover, a substrate bearing two γ-phenyl motifs gave the desired product [18]F]11 in 37% yield. Common halo substituents (F, Cl, Br, and I) were fully compatible with the reaction conditions, showing no competing aryl radiofluorination and providing the desired [18]F-labeled α,α-difluoromethylalkane products, [18]F]18, [18]F]8, [18]F]9, and [18]F]19, respectively, in good yields. Conceivably, such labeled products could serve as intermediates in

**Fig. 3 | Substrate scope for the late-stage radiofluorination of α-bromo-α-fluoromethylalkanes.** Decay-corrected yields are based on HPLC analyses of crude reaction mixtures and expressed as mean ± SD (*n* = 3). Radioactive products were collected at least once for each substrate to verify that HPLC yields matched isolated yields.

multi-stage radiotracer syntheses. α-Bromo-α-fluoroalkanes possessing different heteroaryl rings (naphthyl, thienyl, furanyl, indolyl, and phthalimidyl) readily participated in late-stage radiofluorination to give [18F]**25**, [18F]**16**, [18F]**22**, [18F]**23**, and [18F]**26**, respectively, in yields ranging from 36 to 89%. The method was also remarkably effective across precursors with various alkyl chain lengths, as well as precursors with diverse linker groups like ester ([18F]**18**–[18F]**23**), ether ([18F]**24**, [18F]**25**), and amido ([18F]**26**), as frequently appear in PET tracer candidates[46].

Alcohol, thiol, and amino groups with β-substituted carbons are prevalent in a wide range of drug-like molecules. However, the poor membrane permeability and metabolic vulnerability of such substructures render them quite unattractive for inclusion in tracer designs. The installation of an 18F-labeled α,α-difluoromethyl bioisostere on a branched carbon is a potential strategy to overcome such

limitations[16]. Therefore, having established an efficient protocol for incorporating the 18F-labeled α,α-difluoromethyl group onto a straight alkyl chain, we next turned our attention to examples of more intricate branched aliphatic chain precursors. As a test of feasibility, we subjected several precursors containing a β-secondary carbon attached to the α-bromo-α-fluoromethyl group to late-stage radiofluorination. Gratifyingly, substrates with β-methyl substituents reacted smoothly to generate the desired products, [18F]**27** and [18F]**28**, in 37% and 57% yields, respectively. Additionally, various aliphatic rings, which are among the most common substructures in drug synthesis, were well tolerated in this protocol. Thus, we were able to label the difluoromethyl group with fluorine-18 on an indanyl ring, as in [18F]**29**, in a low but useful 13% yield and on a cyclohexyl ring in a more favorable yield (52%), as in [18F]**30**. Labeled compounds bearing a β-isopropyl group ([18F]**31**, [18F]**32**) were formed in low but useful yields.

**Fig. 4 | Substrate scope for the late-stage radiofluorination of biologically relevant molecules.** Decay-corrected yields are based on HPLC analyses of crude reaction mixtures and expressed as mean ± SD (*n* = 3). Radioactive products were collected at least once for each substrate to verify that HPLC yields matched isolated yields.

## Scope for late-stage modification of biologically relevant molecules

The broad substrate scope found for the late-stage radiofluorination method on model compounds inspired us to further probe the applicability of this method to biologically relevant molecules (Fig. 4). The ¹⁸F-difluoromethyl analogs of vinclozolin, a fungicide ([¹⁸F]**33**), and dicamba, a herbicide ([¹⁸F]**34**) were smoothly synthesized from the corresponding α-bromo-α-fluoro precursors. Moreover, our protocol could be applied to α,α-difluoromethyl analogs of two common non-steroidal anti-inflammatory drugs, ketoprofen ([¹⁸F]**35**) and naproxen ([¹⁸F]**36**), giving yields of 38% and 52%, respectively. Substrates derived from the natural products, nootkatone and isopulegol, were converted into their corresponding ¹⁸F-α,α-difluoromethyl derivatives, [¹⁸F]**37**

and [¹⁸F]**38**, respectively, in good yields. Various ¹⁸F-labeled α,α-difluoromethyl derivatives of nitrogen-containing heterocyclic drugs ([¹⁸F]**39**, [¹⁸F]**40**, and [¹⁸F]**42**), such as an oxazole (oxaprozin), a 1,2,4-oxadiazole (ataluren), and a thiazole (febuxostat), respectively, were obtained in 41–54% yields. Remarkably, the unprotected NH group in the precursor to a derivative of etodolac did not impede radio-fluorination and provided an excellent yield of [¹⁸F]**41** (84%), further demonstrating the compatibility of this methodology with many sensitive functionalities. Complex molecules bearing nitrogen or oxygen linkers, which are widely found in tracers and biologically active compounds, were well-tolerated as shown in the high yields for [¹⁸F] **43**–[¹⁸F]**45**. Two substrates bearing α-bromo-α-fluoro substituents on a branched carbon readily participated in the radiofluorination to give

the [18F]**46** and [18F]**47** in moderate yields. Furthermore, [18F]**45** and [18F]**47**, two derivatives of the well-known COX-1 PET radioligand [11C]PS13[47–49] were readily obtained in 76% and 48% yields, respectively, and exemplify the relevance of this methodology to PET tracer synthesis. Thus, overall this late-stage radiofluorination protocol offers broad opportunity for the straightforward conversion of bioactive molecules into 18F-α,α-difluoromethylalkyl PET imaging tracers.

### Automated syntheses of 18F-labeled α,α-difluoromethylalkanes and the determination of their molar activities

We set out to implement this protocol for radiolabeling difluoromethylalkanes in an automated radiosynthesis module, as would be crucial for clinical application. For that, we chose to automate the synthesis of 18F-labeled α,α-difluoromethylalkanes with a commercially available TracerLab FX2N radiofluorination module. First, we screened several reaction conditions for the optimization of the radiosynthesis of [18F]**10** on the automated radiosynthesis platform. To our delight, treatment of 2.5 μmol (0.73 mg) of precursor **10a** and 2.6 μmol (0.5 mg) of tetraethylammonium bicarbonate with [18F]fluoride (11.5 GBq) gave 1.15 GBq of the corresponding isolated [18F]**10**, as a formulation in sterile saline-ethanol (10: 1 v/v) for injection, with a molar activity of 33 GBq/μmol (entry 1, Supplementary Table 4). Reduction of Et$_4$NHCO$_3$ from 2.6 to 1.3 μmol increased the molar activity of [18F]**10** from 33 to 51 GBq/μmol (entry 1 vs. 2, Supplementary Table 4). Further reduction of the precursor from 2.5 to 1.5 μmol and keeping the base at 1.3 μmol improved the molar activity of [18F]**10** from 51 to 74 GBq/μmol (entry 2 vs. 3, Supplementary Table 4). Thus, reduction of both precursor and base likely reduce fluoride leaching from the precursor, resulting in an increase of molar activity.

Next, we proceeded to show that this methodology was amenable for automation by producing five 18F-labeled model compounds. Starting from modest levels of [18F]fluoride (11–18 GBq), [18F]**10**, [18F]**19**, [18F]**25**, [18F]**36**, and [18F]**39** were obtained in useful non-decay-corrected yields (0.8–1.3 GBq) in 60 to 70 minutes of radiosynthesis time (Fig. 5A). These activity yields exceed what is normally required for PET imaging in a human participant (typically ca 0.55 GBq). Gratifyingly, molar activities ranged from 36 to 100 GBq/μmol and were thus in an acceptable range for PET imaging. Overall, these results demonstrate that this methodology is fully adaptable to automation for PET tracer production.

With respect to measuring molar activity, it is necessary to first remove all non-radioactive byproducts from the radioactive analyte, and especially non-radioactive vinyl products that may elute close to the desired radiolabeled product. Often, the labeled product is readily separable from non-radioactive byproducts by 'semi-preparative' reversed-phase HPLC after some method optimization regarding column type and elution conditions. We demonstrated the feasibility of complete separation of labeled products by single-pass 'semi-preparative' HPLC for [18F]**10**, [18F]**19**, [18F]**25**, [18F]**36**, and [18F]**39** (Supplementary Figs. 14, 18, 22, 26, and 30, respectively). HPLC analytical methods also need to be capable of separating carrier from likely chemical impurities, such as closely eluting elimination products. This was readily shown to be feasible with 3 of 4 tested examples, [18F]**1**, [18F]**7**, and [18F]**13** (Supplementary Figs. 40, 46, and 59).

Finally, to better understand how high product molar activities are achieved with this radiochemical methodology, we conducted several control experiments (Fig. 5B). HPLC analysis of the crude reaction mixture from the radiofluorination of precursor **10a** indicated that the excess α-bromo-α-fluoroalkyl precursor partially undergoes base-mediated HBr elimination to produce a mix of the *E* and *Z* vinyl fluorides, **10b** (Fig. 5B. I and II). However, the only detected radioactive products were the desired [18F]**10** and unreacted [18F]fluoride ion. Neither 18F-labeled **10b** nor 18F-labeled **10a** was formed, showing that isotopic exchange of fluorine-19 for fluorine-18 was not occurring. Chromatographic evidence suggests that HF elimination from **10a** to

generate **10c** is not a viable pathway under the radiofluorination conditions (Supplementary Figs. 34–36). These findings provide strong rationale for the observed high molar activities of the radiofluorinated products. Moreover, the treatment of the vinyl compounds **10b** with [18F]fluoride under the same conditions gave no [18F]**10**, thereby ruling out nucleophilic addition of [18F]fluoride to **10b** as a source of [18F]**10** (Fig. 5B, III). These observations indicate that the dehydrobromination of the α-bromo-α-fluoroalkyl precursor to a non-reactive vinyl fluoride dominates over fluoride release, and that there is also a general absence of isotope exchange as a source of carrier fluoride. A plausible overall mechanism is presented in Supplementary Fig. 37.

In summary, a late-stage radiofluorination method was successfully developed by systematic optimization to produce 18F-labeled α,α-difluoromethylalkyl compounds in moderate to excellent yields and with high molar activities from bromo precursors. The method has broad substrate scope and good functional group tolerance. Furthermore, the radiofluorination works efficiently with both linear and branched alkyl precursors and is useful for installing 18F-labeled α,α-difluoromethyl groups on nitrogen- and oxygen-containing linkers. The method is applicable to labeling complex drug-like molecules as potential PET tracers. These features expand the chemical space that is available for future PET tracer development and especially now enables the special physical and biochemical features of an 18F-labeled α,α-difluoromethyl group to be exploited in PET tracer design.

## Methods

### General procedure for manual evaluation of substrate scope for 18F-radiofluorination

To a 1-mL glass V-vial is added a solution of azeotropically dried [18F]fluoride (50–100 MBq) plus Et$_4$NHCO$_3$ (0.5 mg) in MeCN (200 μL), followed by a solution of substrate (**3, 6a-47a**; 2.5 μmol) in MeCN (300 μL). The reaction vial is maintained with airtightness to prevent the loss of MeCN. The reaction is conducted at a selected temperature for 10 min and quenched with H$_2$O (300 μL). An aliquot is then analyzed using HPLC condition A (see below). Yields is based on HPLC chromatogram peak areas. All yields are decay-corrected and expressed as mean ± SD ($n \geq 2$). Radioactive products are collected at least once for each substrate to verify that HPLC yield matches isolated yield.

HPLC condition A: Luna [C18(2), 250 × 4.6 mm, i.d., Phenomenex], mobile phase: MeCN-50 mM aqueous ammonium formate) at a flow rate of 2 mL/min with MeCN increased from 10% to 50% over 5 min; then further increased to 90% from 5 min to 15 min and kept at 90% MeCN for another 5 min. Eluates are monitored for UV absorbance at 254 nm and for radioactivity.

### General procedure for automated radiosynthesis

The radiosyntheses of the 18F-difluoromethylalkanes ([18F]**10**, [18F]**19**, [18F]**25**, [18F]**36** and [18F]**39**) were performed on a fully automated apparatus (TRACERlab™ FX2 N; GE Healthcare), according to the following procedure. [18F]Fluoride ion (11.1–18.5 GBq) in [18O]water (400–700 μL) and a solution of aqueous Et$_4$NHCO$_3$ (50 μL, 5 mg/mL) are loaded into a glassy carbon vial reactor of the apparatus. MeCN (2.5 mL) is added, and the solvent azeotropically removed at 80–100 °C under a stream of nitrogen gas that is vented to vacuum. This step is repeated after a second addition of MeCN (2.5 mL). The reactor is then cooled to 40 °C and a solution of precursor (**10a, 19a, 25a, 36a** and **39a**; 1.5 μmol) in anhydrous MeCN (0.6 mL) is added. The reaction mixture is heated for 10 min at 120 °C. The reactor is then cooled to 40 °C. The contents are diluted with H$_2$O (2.5 mL), transferred into an intermediate vial, and then delivered to a 5-mL HPLC loop. Under trigger of a fluid detector, the contents are injected onto a semi-preparative HPLC column [Luna C18(2), 250 × 10 mm i.d., 10 μm; Phenomenex], and eluted with a gradient of acetonitrile and 100 mM aqueous ammonium formate at a flow rate of 5 mL/min as described below (HPLC condition C). The eluate is

## A. Automated radiosynthesis

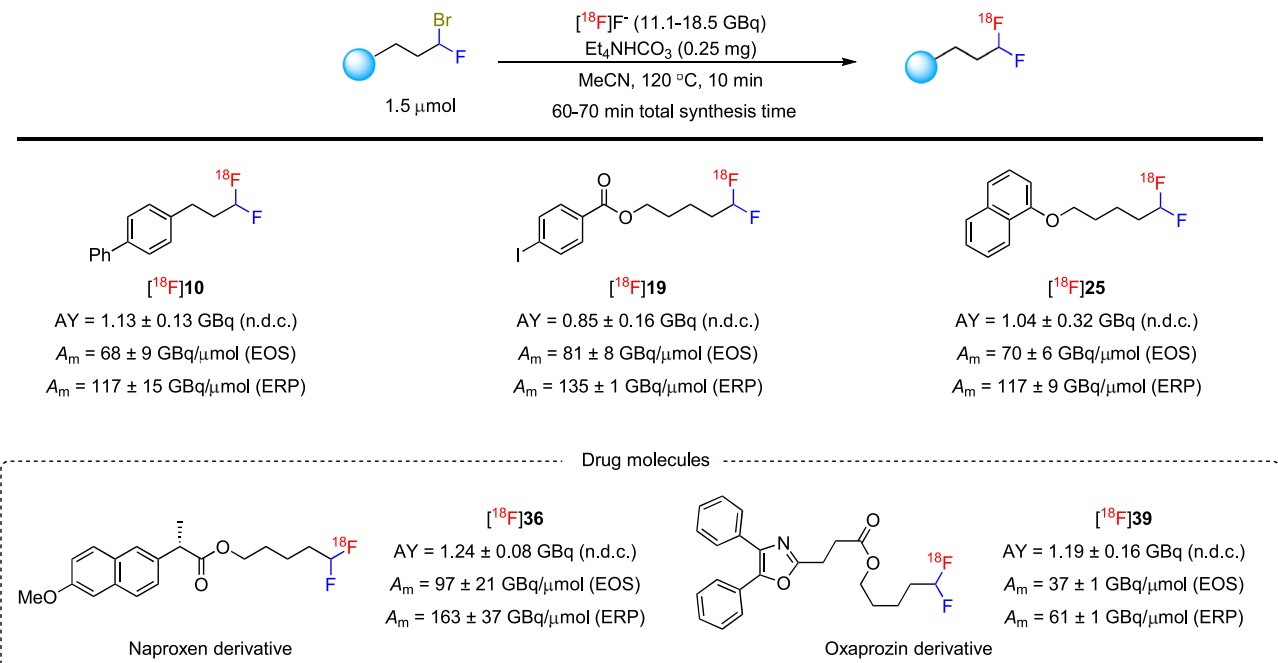

## B. Control experiments

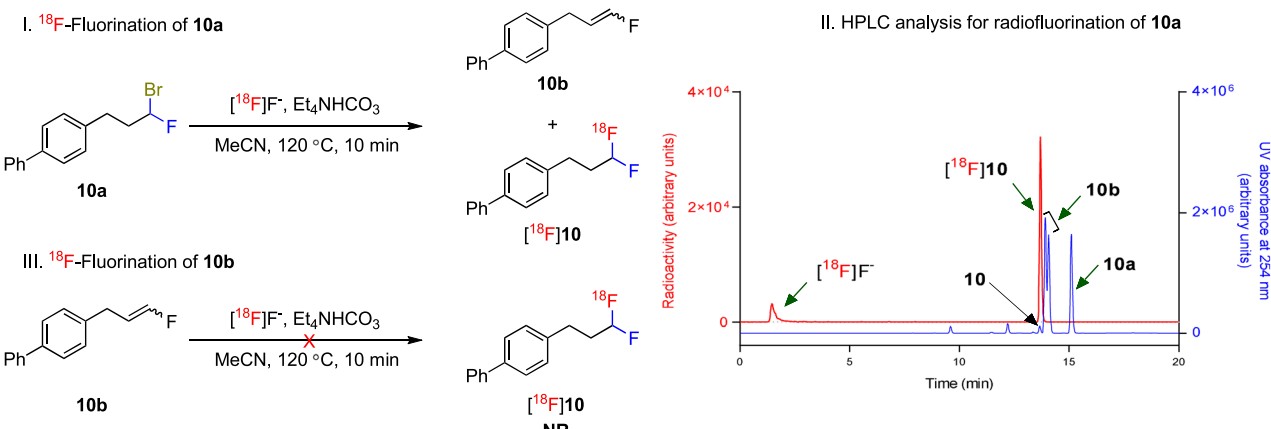

**Fig. 5 | Automated syntheses of $^{18}$F-labeled α,α-difluoromethylalkanes and the determination of their molar activities. A** Automated radiosynthesis. AY activity yield, n.d.c. non-decay corrected, EOS end of radiosynthesis, ERP end of radionuclide production. Yields are mean ± SD for $n = 2$. **B** Control experiments for the synthesis of [$^{18}$F]**10**. NR no reaction.

monitored by first a UV absorbance detector (absorbance at 254 nm) and then a radioactivity detector. The desired product fraction is collected into a round-bottomed flask preloaded with water (40 mL). The contents are then passed through a Sep-Pak plus short C18 cartridge (Waters) preconditioned with ethanol (10 mL) and then water (10 mL). The Sep-Pak cartridge is washed with water (10 mL), and eluted sequentially with ethanol (1 mL) and saline (9 mL) into a final vial. An aliquot of this formulation is used for further analyses, such as determination of radiochemical purity, identity and molar activity.

HPLC condition C: Luna [C18(2), 250 × 10 mm, i.d., Phenomenex], mobile phase: MeCN-100 mM aqueous ammonium formate) at a flow rate of 5 mL/min with a gradient composed of 10% to 50% MeCN over 5 min; further increased to 90% from 5 min to 40 min, and kept at 90% MeCN for another 10 min.

Additional detailed descriptions of experimental, spectroscopic, chromatographic methods and results are given in the Supplementary Information.

## Data availability
The authors declare that the data supporting the findings of this study, including details of materials and methods, NMR spectra, and HPLC chromatograms, are available within the paper and its Supplementary Information files. Any further queries on the data can be directed to either S.T. or V.W.P.

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

## Acknowledgements

We acknowledge the Intramural Research Program of the National Institutes of Health (NIMH; Grant ZIA-MH002793 to V.W.P.) for financial support and thank the NIH Clinical Center PET Department (Chief Dr. P. Herscovitch) for radioisotope production.

## Author contributions

Q.Z. and S.T. contributed to study design and experimental implementation. S.T., S.L., and V.W.P. contributed to study design and supervision. All authors contributed to the writing and review of the manuscript.

## Funding

## Competing interests

The authors declare no competing interests.
