## [Transparent Peer Review file · Nature Communications]

Expanding Tracer Space for Positron Emission Tomography with High Molar Activity ^{18}F -Labeled α,α -Difluoromethylalkanes

Corresponding Author: Dr Victor Pike

Version 0:

Reviewer comments:

Reviewer #1

(Remarks to the Author)

This manuscript by Zhao, et al. reports an improved method for radiolabeling α,α -difluoromethylalkanes by a single-step radiofluorination. This work improves upon the existing technology by enabling the radiolabeling under milder conditions while affording good yields and higher molar activity. The usefulness of the methodology was demonstrated by radiolabeling numerous analogs of existing pharmaceuticals that contain the difluoromethyl group. The ^{18}F -labeled α,α -difluoromethyl group has not been used extensively in the design of PET tracers due to low radiochemical yields and low molar activity. This new methodology now makes it possible to include the α,α -difluoromethyl group in the design of new PET tracers.

Several corrections/concerns are listed below:

1) In the abstract it says ($t_{1/2} = 110$ min) whereas in the first paragraph it says ($t_{1/2} = 109.8$ min). For consistency, these should be the same.

2) Figure 1C says 'chemtype'. Is this supposed to be 'chemotype'?

3) Figure 2B: clarify that 'ND' means 'not detected' rather than 'not determined'.

4) In the fluoride leaching experiments 2, 3, and 4 are reported to have not produced detectable amounts of 1. But 3 and 4 do appear to have produced detectable amounts of the elimination product (compare the ^{19}F NMR resonances centered around -130 ppm in Supplementary Figures 3 and 4 to the ^{19}F NMR resonances of 10b in Supplementary Figure 159). The generation of some elimination product should be acknowledged.

5) Halide leaving groups are generally listed as $\text{F} < \text{Cl} < \text{Br} < \text{I}$ but in Figure 2C $\text{Br} > \text{I}$. Do the authors have any insight into this? A plausible explanation could be that the highly polarizable electron cloud of the iodine atom is drawn toward the carbon atom by the electronegative fluorine atom, thus increasing electron density on the carbon atom and making it less susceptible to nucleophilic attack.

6) Although the fluoride leaching experiments did not indicate that 3 produced 1 in a detectable amount by ^{19}F NMR, the improvement in Am when the amount of precursor was reduced (Supplementary Table 2) suggests that fluoride leaching from the precursor during ^{18}F -18 radiolabeling does happen to some extent. Vinyl fluoride elimination product 10b was synthesized but apparently the vinyl bromide elimination product was not synthesized. The HPLC chromatogram in Figure 5BII ends at 17 minutes. If the more lipophilic vinyl bromide elimination product was present in the reaction mixture, but eluted after 17 minutes, it would not have been detected. The vinyl bromide elimination product should be synthesized and its HPLC retention time determined. If the vinyl bromide elimination product

is present in the reaction mixture, then that would help explain why Am improved when the amount of precursor was reduced.

7) There are errors in the Supplementary Information. A few examples are:

Supplementary Figure 113. ¹H NMR spectrum of compound 1 (400 MHz, CDCl₃)
The chemical structure shown on the spectrum is triflate 5.
This is the same for Supplementary Figures 114 and 115.

Supplementary Figure 116. ¹H NMR spectrum of compound 2 (400 MHz, CDCl₃)
The chemical structure shown on the spectrum is difluoro compound 1.
This is the same for Supplementary Figures 117 and 118.

The errors continue to Supplementary Figure 127.

2.2.8. Specific syntheses

(3,3-Difluoropropyl)benzene (1). General procedure 2 with 3-phenylpropanal followed by flash chromatography (n-hexane) gave 2 as a colorless oil (221 mg, 71% yield).
This should read "gave 1 as a colorless oil".

The Supplementary Information should be checked for any additional errors.

Reviewer #2

(Remarks to the Author)

Reviewer: This manuscript by S. Telu, V. W. Pike and co-workers describes a novel and efficient method based on the single-step radiofluorination of α -bromo- α -fluoroalkanes, which is applicable to bioactive compounds and drug-like molecules in the design of new PET tracers.. This paper is well-written and important related studies are concisely introduced. Therefore, the reviewer wants to support the publication of this paper on Nature Communication.

To improve this manuscript, please address the following points.

- 1) Generally, ¹⁸F-fluorination methods are developed based on well-established ¹⁹F-fluorination techniques and have been thoroughly validated. However, this paper appears to lack a solid experimental foundation for the Br/¹⁹F exchange. To enhance the general significance and applicability of both Br/¹⁹F and ¹⁸F exchanges, it would be beneficial to include additional experiments and discussions of the Br/¹⁹F exchange in this context.
- 2) In Fig.2 C: Radiofluorination of the halo precursors with 4 temperature points, [¹⁸F]1 yield of Br precursor is increased with temperature elevation, such the best 61% at 120 o C, how about higher temperature like 140 and 160 o C? In addition, MeCN as a solvent, higher temperature than its boiling point should have a relationship of pressure and [¹⁸F]1 yield, please investigate it with discussion.
- 3) In Fig.2 B: It is very abrupt to have a discussion of DBU, so please add a simple and graphic description based on the literature 49, so as to read easily.
- 4) In Fig.2 D: Standard condition with ¹⁸F- /Et₄NHCO₃ shows an excellent result, please propose a reasonable mechanism to eluviate if possible, it will be helpful to explore the applications of Br/¹⁹F exchange.
- 5) There are a significant number of English grammatical errors throughout the manuscript which need to be addressed. It should be proofread thoroughly before consideration for publication.

Reviewer #3

(Remarks to the Author)

Ms. Ref. No.: NCOMMS-24-57450-T

Title: Expanding Tracer Space for Positron Emission Tomography: ¹⁸F-Labeled α,α -Difluoromethylalkanes

1. Overview and general recommendation:

In their manuscript, Zhao et al. present a new method for the late-stage radiosynthesis of ¹⁸F-labeled α,α -difluoromethylalkanes via nucleophilic radiofluorination of α -bromo- α -fluoroalkanes with [¹⁸F]fluoride. Through a series of experiments with (3-fluoropropyl)benzene analogs containing different leaving groups on the α -carbon, the authors identified bromo-substituted α -fluoroalkanes as effective precursors for the preparation of α,α -[¹⁸F]difluoromethylalkanes. After optimizing the reaction conditions, the protocol was applied to a diverse range of α -bromo- α -fluoromethylalkanes, showing good compatibility with straight-chain alkyl precursors with various aryl substituents, heteroaryl rings or linker groups. The method also allowed for the radiofluorination of β -branched precursors, though yields were occasionally low. Additionally, the protocol enabled the synthesis of ¹⁸F-difluoromethylated derivatives of 15 drugs and herbicides, with yields ranging from fair to good. Finally, the method was successfully adapted for automated radiosyntheses, enabling the preparation of five ¹⁸F-labeled model compounds with activity yields and molar activities sufficient for imaging applications. Overall, this manuscript is well written, and the authors present their findings in a clear, logical manner that is easy to follow. The methodology is generally rigorous and sound, with the experimental approaches supporting the authors' conclusions

and claims. Given the practical potential of the protocol for PET-chemistry, this study could hold significant interest for the field. However, there are some important aspects that should be addressed before the manuscript will be suitable for publication in Nature Communications. These points are outlined below.

2. Suggested improvements:

2.1 Major points:

2.1.1 Priority issue and appropriate literature citation

In the introduction, the authors only mention previous procedures for the preparation of c.a. α,α -[18F]difluoromethylalkanes (ref. 46), creating the impression that they are the first to prepare n.c.a. α,α -[18F]difluoromethylalkanes. However, this is not the case. In 2019, Frost et al. disclosed two examples for the rapid preparation of n.c.a. α,α -[18F]difluoroalkanes from α,α -[18F]difluoromethylalkenes, which are readily accessible from the corresponding iodonium salts (10.1002/anie.201810413). Additionally, in October 2024, Ortalli et al. published a broad scope method for synthesizing α,α -[18F]difluoromethylalkenes via Mn-mediated oxidative decarboxylation (10.1021/acs.orglett.4c03611). While the latter paper was published after the authors submitted their manuscript, they should revise their title and manuscript text to reflect these prior developments. Specifically, the authors should carefully compare the advantages and limitations of their method with that of Ortalli et al. in terms of aspects like scope, RCYs, molar activities, accessibility of radiolabeling precursors, etc.

The authors further suggest that α,α -[18F]difluoroalkanes may have broader applications than [18F]fluoroalkanes due to reduced in vivo defluorination, citing references 10 and 29–31 to support this claim. However, most (pre)clinical PET-tracers with 18F-substituents at aliphatic positions, such as [18F]FDG, [18F]DPA-714, and [18F]FET, exhibit sufficient in vivo stability. Notably, reference 29 does not address the metabolism of PET-tracers, and according to reference 31 “fluorine-18 present at the terminus of a straight saturated alkyl chain of two or more carbons appears quite resistant to defluorination in human subjects”. Additionally, reference 30 notes that deuteration – a standard approach in PET tracer development – can significantly reduce the defluorination susceptibility of [18F]fluoroalkyl-based tracers. Finally, it is not clear why α,α -difluoroalkanes should generally be more stable than the corresponding fluoroalkanes. The authors should either provide more robust (preferably experimental) evidence to substantiate this point or consider removing it from the manuscript.

2.1.2 Molar activity and chemical purity of the radiolabeled compounds

In the chromatograms provided in the Supporting Information, non-radioactive impurities – possibly the corresponding fluorovinyl derivatives – often appear very close to the product peaks (Rt difference < 0.5 min; difference between UV- and radioactivity signals approximately 0.1 min; see, e.g., 18F-labeled 1, 13–15, 17, 20, 26–28, 33, 35, 36, and 41–43). This proximity should make it extremely difficult to separate the radiolabeled compounds from the side products, potentially affecting both the chemical purity and the apparent molar activity of the isolated products. Despite this, the quality control chromatograms (spiked with non-radioactive reference compounds) consistently display a single UV peak corresponding to the reference compound. To achieve this, did the authors fractionate the product peaks to isolate the desired compounds from side products? For greater clarity, the authors should provide HPLC traces of crude [18F]1, [18F]7, [18F]13 and [18F]14 spiked with the corresponding reference compounds to illustrate the feasibility of separating the products from non-radioactive impurities. If complete separation is difficult to achieve, a discussion of this limitation and its potential impact on purity and molar activity should be included in the manuscript.

2.2 Minor points

2.2.1. Nomenclature consistency

The manuscript would benefit from a thorough revision to ensure adherence to the consensus nomenclature rules for radiopharmaceutical chemistry, as outlined in Coenen et al., Nuclear Medicine and Biology, 2017 [doi: 10.1016/j.nucmedbio.2017.09.004]. In Figures 1–5, “18F–” should be corrected to “[18F]F–” (or “[18F]fluoride”) to align with standard conventions.

2.2.2. Data consistency and error reporting

The statement in the caption of Figure 5B, “Yields are mean \pm range for $n = 2$, except for [18F]10 (mean \pm SD for $n = 3$)” is misleading and appears to be incorrect. According to the Supporting Information (SI; p. S52–53), the data for [18F]10 also seem to be calculated from only two values, rather than three. Additionally, the reported error values diverge from values calculated based on the individual data points provided in the SI. For instance, for [18F]19, the manuscript reports a yield (mean \pm range) of 0.85 ± 0.15 GBq, while my calculations based on the individual values in the SI (0.74 GBq and 0.96 GBq) give a range (maximum - minimum) of 0.22 GBq and a standard deviation (SD) of 0.16 GBq. Similarly, for [18F]25, the reported yield is 1.04 ± 0.33 GBq, but calculations based on the SI values (1.26 GBq and 0.81 GBq) result in a range of 0.45 GBq and an SD of 0.32 GBq.

In these cases, the error values reported in the manuscript align more closely with the standard deviation than with the range, though the precise basis for these values remains unclear. Additionally, for [18F]10, [18F]36 and [18F]39, there are slight discrepancies between the reported means (1.15, 1.26 and 1.18) and means calculated based on the SI data (1.13, 1.24 and 1.19), potentially due to rounding differences.

To improve clarity, I recommend revising the figure caption (or addressing this issue elsewhere in the manuscript) to specify whether the error values represent standard deviation, range, or another measure. Additionally, verifying that all reported mean values align with the SI data and clearly stating any rounding methods applied would ensure consistency and transparency in data reporting.

2.2.3. Figs. 4 and 5 and text corrections

In the reaction scheme of Fig. 4, please remove the term “New candidate tracer”. Additionally, in Figs. 4 and 5, as well as in the corresponding sections of the manuscript, replace the term “analogues” with “derivatives”.

2.2.4 Determination of molar activity

Please provide additional details in the Supporting Information regarding the determination of molar activity. Additionally, include data points for the experimental carrier amount on the calibration curves to enhance clarity and reproducibility

Version 1:

Reviewer comments:

Reviewer #1

(Remarks to the Author)

The requested changes have been made. This has improved the overall quality of the manuscript and it is suitable for publication.

Reviewer #3

(Remarks to the Author)

The authors have carefully assigned all concerns expressed in the initial review. Accordingly, I am happy to recommend the revised paper for publication in the Journal.

Re: manuscript NCOMMS-24-57450-T

Here follows our response to the reviewers. Matters suggesting attention by reviewers are highlighted in yellow; our responses to reviewers are in blue type, and the revision material in the Manuscript and the Supplementary Information is in red type.

Reviewer # 1 (Remarks to the Author):

This manuscript by Zhao, et al. reports an improved method for radiolabeling α,α -difluoromethylalkanes by a single-step radiofluorination. This work improves upon the existing technology by enabling the radiolabeling under milder conditions while affording good yields and higher molar activity. The usefulness of the methodology was demonstrated by radiolabeling numerous analogs of existing pharmaceuticals that contain the difluoromethyl group. The ^{18}F -labeled α,α -difluoromethyl group has not been used extensively in the design of PET tracers due to low radiochemical yields and low molar activity. This new methodology now makes it possible to include the α,α -difluoromethyl group in the design of new PET tracers.

We sincerely thank Reviewer 1 for the positive review and for valuable suggestions to improve the presentation of the scientific impact of this work. We also appreciate Reviewer 1's meticulous proofreading of both the Manuscript and Supplementary Information for disclosure of typos and minor errors.

Several corrections/concerns are listed below:

Comments:

1) In the abstract it says ($t_{1/2} = 110$ min) whereas in the first paragraph it says ($t_{1/2} = 109.8$ min). For consistency, these should be the same.

$t_{1/2} = 109.8$ min (the most widely accepted value) is now used in the abstract and throughout the manuscript. The abstract now says "Cyclotron-produced short-lived fluorine-18 ($t_{1/2} = 109.8$ min) is widely used to radiolabel tracers for PET."

2) Figure 1C says 'chemtype'. Is this supposed to be 'chemotype'?

We fixed this typo in the new version of Fig. 1C (Page 2).

3) Figure 2B: clarify that 'ND' means 'not detected' rather than 'not determined'.

Fig. 2B legend (Page 4) is now revised to read “(B) Study of fluoride leaching from precursors 1–5 under typical radiofluorination conditions but in the absence of added fluoride. Yields were determined by ¹⁹F-NMR analysis of the crude reaction mixtures. N/A = not applicable; ND = not detected.”

4) In the fluoride leaching experiments 2, 3, and 4 are reported to have not produced detectable amounts of 1. But 3 and 4 do appear to have produced detectable amounts of the elimination product (compare the ¹⁹F NMR resonances centered around -130 ppm in Supplementary Figures 3 and 4 to the ¹⁹F NMR resonances of 10b in Supplementary Figure 159). The generation of some elimination product should be acknowledged.

We now acknowledge the generation of elimination products from precursors 3 and 4 by adding the following to the Results and Discussions section (Page 5) when explaining the results of **Fig. 2B**. “The halo precursors 3 (X = Br) and 4 (X = I) however did appear to produce some fluorine-containing elimination product (as an *E/Z* isomer mixture), as evidenced by ¹⁹F-NMR resonances at -129.6 and -131.6 ppm (Supplementary Figures 3 and 4).”

We revised **Supplementary Figures 3 and 4** and clearly labeled the elimination product peaks in the ¹⁹F NMR spectra. Additionally, we added “ND = not detected” to the legends of **Supplementary Figures 3 and 4**.

Supplementary Figure 3. Fluoride leaching experiment with 3 (X = Br). ND = not detected.

Supplementary Figure 4. Fluoride leaching experiment with **4** (X = I). **ND = not detected.**

5) Halide leaving groups are generally listed as $\text{F} < \text{Cl} < \text{Br} < \text{I}$. Do the authors have any insight into this? A plausible explanation could be that the highly polarizable electron cloud of the iodine atom is drawn toward the carbon atom by the electronegative fluorine atom, thus increasing electron density on the carbon atom and making it less susceptible to nucleophilic attack.

We thank Reviewer 1 for this helpful comment and question. Therefore, in the manuscript where we discuss the results of **Fig. 2C** (Page 6), we now add a plausible explanation for the observed lower reactivity of the iodo precursor (**4**) than the bromo precursor (**3**), as follows:

“The greater reactivity of the bromo precursor **3** over that of the iodo precursor **4** deserves comment. The leaving group abilities of halo substituents in aliphatic nucleophilic substitution reactions normally follow the order $\text{I} > \text{Br} > \text{Cl} > \text{F}$. The observed higher performance of the bromo precursor (**3**) over the iodo precursor (**4**) could be because the electron cloud of the iodo substituent is more polarizable than that of the bromo substituent. As a result, the electron cloud on iodine may be more easily drawn towards the carbon by the electronegative α -fluorine substituent, thereby increasing the electron density on the carbon center and making it less susceptible to nucleophilic attack by ^{18}F fluoride ion.”

6) Although the fluoride leaching experiments did not indicate that 3 produced 1 in a detectable amount by ^{19}F NMR, the improvement in Am when the amount of precursor was reduced (Supplementary Table 2) suggests that fluoride leaching from the precursor during F-18 radiolabeling does happen to some extent.

Indeed, both amounts of precursor and base may impact the molar activity of the ^{18}F -labeled products. We added the following comment to the manuscript (Page 11) to discuss this possibility and to clarify the observations in the now renumbered **Supplemental Table 4**.

“Reduction of Et_4NHCO_3 from 2.6 to 1.3 μmol increased the molar activity of $[^{18}\text{F}]\mathbf{10}$ from 33 to 51 GBq/ μmol (entry 1 vs. 2, Supplementary Table 4). Further reduction of the precursor from 2.5 to 1.5 μmol and keeping the base at 1.3 μmol improved the molar activity of $[^{18}\text{F}]\mathbf{10}$ from 51 to 74 GBq/ μmol (entry 2 vs. 3, Supplementary Table 4). Thus, reduction of both precursor and base likely reduce fluoride leaching from the precursor, resulting in an increase of molar activity.”

Vinyl fluoride elimination product 10b was synthesized but apparently the vinyl bromide elimination product was not synthesized. The HPLC chromatogram in Figure 5BII ends at 17 minutes. If the more lipophilic vinyl bromide elimination product was present in the reaction mixture, but eluted after 17 minutes, it would not have been detected. The vinyl bromide elimination product should be synthesized and its HPLC retention time determined. If the vinyl bromide elimination product is present in the reaction mixture, then that would help explain why Am improved when the amount of precursor was reduced.

We appreciate this insightful suggestion. We agreed with Reviewer 1 that the suggested control experiment would clarify the influence of potential HF elimination on the radiofluorination and on the molar activity of the radiofluorination products. Therefore, we synthesized the vinyl bromide (**10c**) as a mixture of *E/Z* isomers according to a literature procedure, as shown below. **10c** was fully characterized by NMR and HRMS. The following text is now included in the Supplementary Information.

Details of the synthesis of **10c** and the characterization data of **10c** are shown at **Supplementary Information** pages **S18** and **S19**. The ^1H -NMR and ^{13}C -NMR spectra are listed as **Supplementary Figures 179** and **180**.

(*E/Z*)-4-(3-bromoallyl)-1,1'-biphenyl (10c)¹⁰.

To a flame-dried flask was added 2-([1,1'-biphenyl]-4-yl)acetaldehyde (589 mg, 3.0 mmol, 1.0 equiv.), CBr_4 (1.49 g, 4.5 mmol, 1.5 equiv.), and CH_2Cl_2 (10 mL). The flask was cooled to 0 °C, and then a solution of PPh_3 (2.36 g, 9.0 mmol, 3.0 equiv.) in CH_2Cl_2 (5.0 mL) was added dropwise from a syringe over 10 min. The solution was stirred at 0 °C under N_2 for 1 h. About half of the volume of CH_2Cl_2 was removed under reduced pressure. Hexane (50 mL) was added. Triphenylphosphine oxide (TPPO) precipitated out. After filtration and evaporation of the filtrant, the residue was dissolved in hexane (50 mL) which led to further precipitation of TPPO. Filtration and evaporation of the solvent afforded crude 4-(3,3-dibromoallyl)-1,1'-biphenyl which was used directly in the next step, as follows.

To a solution of the crude 4-(3,3-dibromoallyl)-1,1'-biphenyl (~ 2.0 mmol, 1.0 equiv.) and NEt_3 (607 mg, 6.0 mmol, 3.0 equiv.) in DMF (3.0 mL) was added dimethyl phosphonate (660 mg, 6.0 mmol, 3.0 equiv.). The solution was stirred overnight at room temperature. Water (10 mL) was added to the mixture, which was then extracted with diethyl ether (50 mL). The combined organic phase was separated, washed with brine, dried (Mg_2SO_4), and concentrated under vacuum. The residue was purified by silica gel flash column chromatography (*n*-hexane) to afford **10c** as a white solid (293 mg, E/Z = 8/5, 36% yield). Mp: 42–44 °C.

^1H NMR (400 MHz, CDCl_3) δ (ppm): 7.69–7.55 (m, 4H), 7.51–7.47 (m, 2H), 7.45–7.27 (m, 3H), 6.56–6.30 (m, 1.33H), 6.18 (dt, J = 13.5, 1.4 Hz, 0.58H), 3.84–3.55 (m, 0.79H), 3.45 (d, J = 7.1 Hz, 1.24H); ^{13}C NMR (101 MHz, CDCl_3) δ (ppm): 141.08, 141.00, 139.73, 139.60, 137.95, 137.46, 136.63, 133.58, 129.12, 129.04, 128.96, 128.94, 127.54, 127.53, 127.41, 127.35, 127.21, 108.98, 106.21, 38.91, 35.83. MS (EI) m/z : 272.0 [M]⁺. HRMS (EI) m/z for $\text{C}_{15}\text{H}_{13}\text{Br}$ [M]⁺: calc'd: 272.0201; found: 272.0201.

Supplementary Figure 179. ^1H NMR spectrum of compound **10c** (400 MHz, CDCl_3)

Supplementary Figure 180. ^{13}C NMR spectrum of compound **10c** (101 MHz, CDCl_3)

We conducted the radiofluorination of precursor **10a** as a model substrate and monitored the reaction products with HPLC to determine whether the vinyl bromides (**10c**) form during the reaction. The retention times of **10c** (a mixture of *E/Z* isomers) were 15.4 and 15.6 min under conditions identical to those used for the analysis of the crude reaction mixture from the radiofluorination of **10a** (**Supplementary Figure 34**). We extended the HPLC analysis time to 20 min, as suggested by Reviewer 1, and as now shown in **Supplementary Figure 35**. Subsequently, we co-injected the crude mixture from the radiofluorination of the precursor **10a** with the mixture of vinyl bromides **10c** (**Supplementary Figure 36**). Absence of the **10c** peaks in the HPLC chromatogram of the crude radiofluorination reaction mixture (**Supplementary Figure 35**), as compared to that of the co-injection sample (**Supplementary Figure 36**), indicates precursor **10a** did not generate **10c**, an HF elimination product, in detectable amounts under the radiochemical conditions. This data strongly supports one of the key aspects of this work, i.e. that our method is effective for giving high molar activity [¹⁸F]α,α-difluoromethyl alkanes. The HPLC data are summarized in Section 3.4.2. of the Supplementary Information.

3.4.2. Investigation of the potential formation of the vinyl bromide **10c** in the radiofluorination of **10a**

Supplementary Figure 34. Analytical HPLC chromatogram (condition A) for reference compound **10c** (*E* and *Z* isomer mixture).

Supplementary Figure 35. Analytical HPLC chromatogram (condition A) for unpurified $[^{18}\text{F}]\mathbf{10}$ produced from $\mathbf{10a}$.

Supplementary Figure 36. Analytical HPLC chromatogram (condition A) for co-injection of reaction mixture producing $[^{18}\text{F}]\mathbf{10}$ from $\mathbf{10a}$ and reference compound $\mathbf{10c}$ (*E* and *Z* isomer mixture).

The following text was inserted into the manuscript (Page 11).

“Chromatographic evidence suggests that HF elimination from $\mathbf{10a}$ to generate $\mathbf{10c}$ is not a viable pathway under the radiofluorination conditions (Supplementary Figures 34–36). These findings provide strong rationale for the observed high molar activities of the radiofluorinated products.”

We also revised **Fig. 5B. II** (Page 10) by extending the HPLC elution time to 20 min to ensure that no other side-products, such as non-radioactive $\mathbf{10c}$, were obscured, as shown below:

7) There are errors in the Supplementary Information. A few examples are:

Supplementary Figure 113. ^1H NMR spectrum of compound 1 (400 MHz, CDCl_3)

The chemical structure shown on the spectrum is triflate 5.

This is the same for Supplementary Figures 114 and 115.

Supplementary Figure 116. ^1H NMR spectrum of compound 2 (400 MHz, CDCl_3)

The chemical structure shown on the spectrum is difluoro compound 1.

This is the same for Supplementary Figures 117 and 118.

The errors continue to Supplementary Figure 127.

We thank Reviewer 1 for meticulous attention to the details. We corrected all the compound numbers and structures, so that they match those in the following Supplementary Figures.

We corrected the chemical structure of **1** in **Supplementary Figures 113–115** (now re-numbered **Supplementary Figures 131–133**).

We corrected the chemical structure of **2** in **Supplementary Figures 116–118** (now re-numbered as **Supplementary Figures 134–136**).

We corrected the chemical structure of **3** in **Supplementary Figures 119–121** (now re-numbered as **Supplementary Figures 137–139**).

We corrected the chemical structure of **4** in **Supplementary Figures 122–124** (now re-numbered as **Supplementary Figures 140–142**).

We corrected the chemical structure of **5** in **Supplementary Figures 125–127** (now re-numbered as **Supplementary Figures 143–145**).

2.2.8. Specific syntheses

(3,3-Difluoropropyl)benzene (1). General procedure 2 with 3-phenylpropanal followed by flash chromatography (n-hexane) gave 2 as a colorless oil (221 mg, 71% yield).

This should read “gave 1 as a colorless oil”.

Correction done at section 2.4.8 (previously 2.2.8) of Supporting Information (Page S13) as follows: “General procedure 2 with 3-phenylpropanal followed by flash chromatography (*n*-hexane) gave 1 as a colorless oil (221 mg, 71% yield)”.

The Supplementary Information should be checked for any additional errors.

We carefully checked the Supplementary Information for additional errors and made appropriate corrections.

Reviewer #2 (Remarks to the Author):

Reviewer: This manuscript by S. Telu, V. W. Pike and co-workers describes a novel and efficient method based on the single-step radiofluorination of α -bromo- α -fluoroalkanes, which is applicable to bioactive compounds and drug-like molecules in the design of new PET tracers. This paper is well-written and important related studies are concisely introduced. Therefore, the reviewer wants to support the publication of this paper on Nature Communication.

We appreciate the positive comments on the manuscript, and we are grateful for the support to publish this work in Nature Communications We also thank Reviewer 2 for insightful suggestions on mechanistic studies.

To improve this manuscript, please address the following points.

1) Generally, ^{18}F -fluorination methods are developed based on well-established ^{19}F -fluorination techniques and have been thoroughly validated. However, this paper appears to lack a solid experimental foundation for the Br/ ^{19}F exchange. To enhance the general significance and applicability of both Br/ ^{19}F and ^{18}F exchanges, it would be beneficial to include additional experiments and discussions of the Br/ ^{19}F exchange in this context.

We appreciate the Reviewer's important suggestion. We took up the suggestion by carrying out Br/¹⁹F⁻ exchange experiments with bromo precursor **3** and tetramethylammonium fluoride as fluoride source. Tetramethyl ammonium fluoride was used as the limiting reagent in the reactions. Yields of Br/¹⁹F⁻ exchange product **1** and elimination products **1b** were determined with ¹⁹F-NMR using (1,1,1-trifluoromethyl)benzene (δ -62 ppm) as internal standard. The conditions that were closer to those of the radiofluorination reaction (i.e. using a large excess of bromo precursor with regard to the fluoride ion), provided the desired fluoro product **1** (δ -116.4 ppm) in useful yields along with the elimination products *E/Z-1b* (δ -129.58 and -131.60 ppm). Increasing the number of fluoride equivalents with respect to bromo precursor neither improved the yield of **1** nor altered the product composition between **1** and *E/Z-1b*. If no external fluoride was added, only the unreacted precursor **3** was detected as shown by single ¹⁹F-NMR resonance at δ -131.1 ppm. Absence of ¹⁹F-NMR resonances at δ -116.4, -129.58, and -131.60 ppm demonstrated the precursor's stability and provided evidence for no detectable fluoride leaching. We have commented and provided an explanation in the Results and Discussion (page 5) of the manuscript, as follows:

"We therefore focused our further investigations on the halo precursors, 2–4.

We next explored reaction stoichiometry using **3** as a model substrate. Reactions of **3** (0.05 mmol) in acetonitrile at 120 °C for 10 min, with the number of equivalents of tetramethylammonium fluoride varied between zero and one, were monitored by ¹⁹F-NMR using (1,1,1-trifluoromethyl)benzene as an internal standard. With no externally added fluoride, only the unreacted precursor **3** was detected by ¹⁹F-NMR (δ -131.1 ppm) (Supplementary Figure 8). Otherwise, irrespective of the number of fluoride equivalents, the product consisted of about 10% of the desired α,α -difluoromethylalkane **1** (δ -116.4 ppm) and 25–30% of the elimination products *E/Z-1b* (δ -129.58 and -131.60 ppm) (Supplementary Table 1 and Supplementary Figures 9–11). These results indicated that substitution of a higher halo substituent was likely a viable route for producing NCA ¹⁸F-labeled α,α -difluoromethylalkanes from [¹⁸F]fluoride, where the precursor would be in very large molar excess over [¹⁸F]fluoride, but not a practical route for producing non-radioactive α,α -difluoromethylalkanes in high yields."

These experiments undoubtedly strengthen the scientific content of the manuscript and provide an adequate foundation for the radiofluorination of α -bromo- α -fluoroalkane precursors for accessing high molar activity [¹⁸F] α,α -difluoroalkanes.

A new section **"2.3. ¹⁹F exchange experiments"** is added to Supplementary Information, which describes the experimental procedure and parameters. The results are summarized in **Supplementary Table 1**; the ¹⁹F-NMR data are shown in **Supplementary Figures 8–11**.

2.3. ^{19}F exchange experiments

Procedure:

Precursor (3-bromo-3-fluoropropyl)benzene (**3**, 0.05 mmol), tetramethylammonium fluoride (x mmol) and MeCN (0.5 mL) were added to a 1-mL glass vial equipped with a magnetic stirrer bar. The vial was capped loosely and then the contents were stirred at 120 °C for 10 min. The reaction was then quenched with water (0.2 mL). The yields of (3,3-difluoropropyl)benzene (**1**, δ -116.4 ppm) and (*E/Z*)-(3-fluoroallyl)benzene (**1b**, δ -129.6 and -131.6 ppm) were determined based on the amounts of tetramethylammonium fluoride by ^{19}F -NMR analysis of the crude reaction mixture with PhCF_3 (δ -62.0 ppm) as internal standard. The results and ^{19}F -NMR spectra are shown below.

Results:

Supplementary Table 1. ^{19}F exchange experiments examining the effect of precursor **3** to Me_4NF ratio on the formation of substitution product **1** and elimination product **1b**.

Reaction scheme showing the conversion of precursor **3** (3-bromo-3-fluoropropyl)benzene to products **1** (3,3-difluoropropyl)benzene and **1b** (*E/Z*)-(3-fluoroallyl)benzene. The reaction conditions are MeCN, 120 °C, 10 min, and Me_4NF (x mmol). Precursor **3** is 0.05 mmol.

Entry	x	Ratio (3 / Me_4NF)	Yield of 1 (%)	Yield of 1b (%)
1	0.05	1 / 1	9	29
2	0.025	2 / 1	11	26
3	0.005	10 / 1	9	31
4	0	-	0	0

Supplementary Figure 8. ¹⁹F exchange experiment of **3** in the absence of Me₄NF.

Supplementary Figure 9. ¹⁹F exchange experiment of **3** with Me₄NF (0.05 mmol, 1.0 equiv.).

Supplementary Figure 10. ¹⁹F exchange experiment of **3** with Me₄NF (0.025 mmol, 0.5 equiv.).

Supplementary Figure 11. ¹⁹F exchange experiment of **3** with Me₄NF (0.005 mmol, 0.1 equiv.).

2) In Fig.2 C: Radiofluorination of the halo precursors with 4 temperature points, [^{18}F]1 yield of Br precursor is increased with temperature elevation, such the best 61% at 120 o C, how about higher temperature like 140 and 160 o C? In addition, MeCN as a solvent, higher temperature than its boiling point should have a relationship of pressure and [^{18}F]1 yield, please investigate it with discussion.

As Reviewer 2 recommended, we performed extra radiofluorination experiments at the higher temperatures of 140 and 160 °C with the bromo precursor **3**. We did not observe substantial beneficial effect at these higher temperatures on the yield of [^{18}F]1 beyond that at 120 °C. The data are summarized in the updated **Fig. 2C** and **Supplementary Table 2**. The suggested investigation of the relationship between pressure and the yield of [^{18}F]1 at higher temperature is not possible due to practical limitations imposed by the reactor and considerations for personnel radiation safety.

Updated **Figure 2C**:

Updated **Supplementary Table 2**:

Supplementary Table 2. Radiofluorination of different precursors at different temperatures.

T (°C)	Yield (%)			
	X = F	X = Cl	X = Br	X = I
40	0	trace	5 ± 1	6 ± 1
60	0	trace	19 ± 2	13 ± 1
80	0	1 ± 0	34 ± 4	18 ± 3
100	0	1 ± 0	41 ± 2	19 ± 3
120	0	7 ± 3	61 ± 8	20 ± 1
140	-	-	62 ± 1	-
160	-	-	64 ± 2	-

In the Results and Discussion section of the manuscript where **Fig. 2C** results are discussed (Page 6), we added:

“For further radiofluorination optimization, we focused on the high-performing bromo precursor **3**. We explored the effect of higher temperature. However, the yield of [¹⁸F]**1** did not increase substantially between 120 °C and 160 °C (Fig. 2C and Supplementary Table 2). Thus, further optimizations, looking at base, solvent, and amount of precursor, were performed at 120 °C to keep the reaction conditions relatively mild, especially when using low boiling-point acetonitrile as solvent (Fig. 2D).”

3) In Fig.2 B: It is very abrupt to have a discussion of DBU, so please add a simple and graphic description based on the literature 49, so as to read easily.

A graphical description to illustrate how DBU mediates the intermolecular fluorination of 1-fluorononotriflate is now added as **Supplementary Figure 7**.

2.2. Proposed mechanism for reaction of DBU with 1-fluorononyl triflate¹

Supplementary Figure 7. Proposed mechanism for the reaction of DBU with 1-fluorononyl triflate.

We modified the text in the manuscript (Page 5), as follows:

“Our finding is consistent with a report that the treatment of 1-fluorononyl triflate with the base, 1,8-diazabicyclo[5.4.0]undec-7-ene (DBU), slowly produces α,α -difluorononane because DBU extracts fluoride ion from the substrate by an S_N2 reaction; this fluoride then rapidly displaces triflate from another molecule of 1-fluorononyl triflate, as depicted in Supplementary Figure 7⁴³.”

4) In Fig.2 D: Standard condition with $^{18}\text{F}^-/\text{Et}_4\text{NHCO}_3$ shows an excellent result, please propose a reasonable mechanism to eluviate if possible, it will be helpful to explore the applications of Br/ ^{19}F exchange.

We propose a plausible mechanism for the radiofluorination reaction using precursor **10a** as a model compound (**Supplementary Figure 37**) for the formation of product via an S_N2 mechanism (i.e., ^{18}F **10**), based on the observations reported in Fig. 5). In **Supplementary Figure 37**, we also include a competing elimination pathway which produces non-radioactive **10b**.

3.4.3. Proposed mechanism for the formation of α,α -difluoromethylalkane [^{18}F]**10**

Supplementary Figure 37. Proposed mechanism for the radiofluorination α -fluoro, α -bromo alkane, exemplified with **10a**.

We modified the discussion of Fig. 5B II (Page 11) to include the experimental results (**Supplementary Figures 34–36**) obtained for this revision, as follows:

“Moreover, the treatment of the vinyl compounds **10b** with [¹⁸F]fluoride under the same conditions gave no [¹⁸F]**10**, thereby ruling out nucleophilic addition of [¹⁸F]fluoride to **10b** as a source of [¹⁸F]**10** (Fig. 5B, III). These observations indicate that the dehydrobromination of the α -bromo- α -fluoroalkyl precursor to a non-reactive vinyl fluoride dominates over fluoride release, and that there is also a general absence of isotope exchange as a source of carrier fluoride. A plausible overall mechanism is presented in Supplementary Figure 37.”

5) There are a significant number of English grammatical errors throughout the manuscript which need to be addressed. It should be proofread thoroughly before consideration for publication.

All authors carefully proofread the entire manuscript and corrected several grammatical errors. Furthermore, we examined the manuscript with a grammar checking program (<https://quillbot.com/grammar-check>) which revealed numerous but mainly small errors that we then corrected.

Reviewer #3 (Remarks to the Author):

Ms. Ref. No.: NCOMMS-24-57450-T

Title: Expanding Tracer Space for Positron Emission Tomography: ¹⁸F-Labeled α,α -Difluoromethylalkanes

1. Overview and general recommendation:

In their manuscript, Zhao et al. present a new method for the late-stage radiosynthesis of ¹⁸F-labeled α,α -difluoromethylalkanes via nucleophilic radiofluorination of α -bromo- α -fluoroalkanes with [¹⁸F]fluoride. Through a series of experiments with (3-fluoropropyl)benzene analogs containing different leaving groups on the α -carbon, the authors identified bromo-substituted α -fluoroalkanes as effective precursors for the preparation of α,α -[¹⁸F]difluoromethylalkanes. After optimizing the reaction conditions, the protocol was applied to a diverse range of α -bromo- α -fluoromethylalkanes, showing good compatibility with straight-chain alkyl precursors with various aryl substituents, heteroaryl rings or linker groups. The method also allowed for the radiofluorination of β -branched precursors, though yields were occasionally low. Additionally, the protocol enabled the synthesis of ¹⁸F-difluoromethylated derivatives of 15 drugs and

herbicides, with yields ranging from fair to good. Finally, the method was successfully adapted for automated radiosyntheses, enabling the preparation of five ^{18}F -labeled model compounds with activity yields and molar activities sufficient for imaging applications. Overall, this manuscript is well written, and the authors present their findings in a clear, logical manner that is easy to follow. The methodology is generally rigorous and sound, with the experimental approaches supporting the authors' conclusions and claims. Given the practical potential of the protocol for PET-chemistry, this study could hold significant interest for the field. However, there are some important aspects that should be addressed before the manuscript will be suitable for publication in Nature Communications. These points are outlined below.

We thank Reviewer 3 for the meticulous review of our Manuscript and Supplementary Information, as well as for recognizing the significant impact that this work could have on the radiochemistry field, and thus recommending publication of this work in *Nature Communications*. We also appreciate this reviewer for the suggestions on literature references, reporting of data, insights into HPLC purifications, especially with respect to potential practical applications, and finally nomenclature.

2. Suggested improvements:

2.1 Major points:

2.1.1 Priority issue and appropriate literature citation

In the introduction, the authors only mention previous procedures for the preparation of c.a. α,α - ^{18}F difluoromethylalkanes (ref. 46), creating the impression that they are the first to prepare n.c.a. α,α - ^{18}F difluoromethylalkanes. However, this is not the case. In 2019, Frost et al. disclosed two examples for the rapid preparation of n.c.a. α,α - ^{18}F difluoroalkanes from α,α - ^{18}F difluoromethylalkenes, which are readily accessible from the corresponding iodonium salts (10.1002/anie.201810413). Additionally, in October 2024, Ortalli et al. published a broad scope method for synthesizing α,α - ^{18}F difluoromethylalkenes via Mn-mediated oxidative decarboxylation (10.1021/acs.orglett.4c03611). While the latter paper was published after the authors submitted their manuscript, they should revise their title and manuscript text to reflect these prior developments. Specifically, the authors should carefully compare the advantages and limitations of their method with that of Ortalli et al. in terms of aspects like scope, RCYs, molar activities, accessibility of radiolabeling precursors, etc.

We agree with reviewer's suggestions. The title has been revised to "Expanding Tracer Space for Positron Emission Tomography: High Molar Activity ^{18}F -Labeled α,α -Difluoromethylalkanes" to strength the message of the key findings and advances in this work.

We also thank the Reviewer for catching our lapses in citation of the latest literature. The suggested two papers are now cited:

39. Frost, A. B., Brambilla, M., Exner, R. M. & Tredwell, M. Synthesis and derivatization of 1,1- ^{18}F difluorinated alkenes. *Angew. Chem. Int. Ed.* **58**, 472-476 (2019).
40. Ortalli, S. et al. ^{18}F -Difluoromethyl(ene) motifs via oxidative fluorodecarboxylation with ^{18}F fluoride. *Org. Lett.* **26**, 9368-9372 (2024).

We modified **Fig. 1B** (Prior work) (Page 2) as well as the Figure legend to reflect the three literature methods for access ^{18}F -Labeled α,α -difluoromethylalkanes, giving the new three new references mentioned in the preceding paragraph.

Updated Figure 1B:

B. Prior work

Fig. 1 | Biomolecules containing α,α -difluoromethyl groups and approaches for their labeling with fluorine-18. (A) Examples of biomolecules containing α,α -difluoromethylalkyl groups. (B) Prior methods for accessing ^{18}F -labeled α,α -difluoromethylalkyl groups and their limitations. (C) Novel methodology for synthesizing ^{18}F -labeled α,α -difluoromethylalkyl groups at high molar activity, as presented in this work.

In the Introduction, we now insert a brief comment comparing advantages of our method over literature methods, with emphasis on our method's overall robustness, such as straightforward synthesis, not requiring transition metals, late-stage (i.e. single step) labeling, broad substrate scope, and most importantly providing PET radiotracers with high molar activity. Another major impact of our method is that it can easily be translated to automated radiosynthesis platforms to

facilitate routine PET radiotracer production for practical applications in the field. The text in the revised manuscript (Page 3), now communicates this more clearly and reads:

“To date, few methods exist for the radiosynthesis of an ^{18}F -labeled α,α -difluoromethylalkane. The earliest report showed only one example with a low yield from an α -fluoroalkyl aryl thioether precursor by the combined action of the oxidant, 1,3-dibromo-5,5-dimethylhydantoin, and a large excess of carrier-added pyridine-9H[^{18}F]F 38 (Fig. 1B). More recently two methods have appeared^{39,40}. One method requires two steps from an α -aryldifluorovinyl iodonium salt and gives a low molar activity³⁹. Another method is single-step from a 2-fluoroalkanoic acid precursor but gives only somewhat improved molar activity⁴⁰. This method also requires the use of an oxidizing agent and a metal-ligand mediator. Thus, there is a clear need for the development of simple and efficient broad-scope methodologies for labeling α,α -difluoromethylalkanes with fluorine-18 at high molar activity.”

The authors further suggest that α,α -[^{18}F]difluoroalkanes may have broader applications than [^{18}F]fluoroalkanes due to reduced *in vivo* defluorination, citing references 10 and 29–31 to support this claim. However, most (pre)clinical PET-tracers with ^{18}F -substituents at aliphatic positions, such as [^{18}F]FDG, [^{18}F]DPA-714, and [^{18}F]FET, exhibit sufficient *in vivo* stability. Notably, reference 29 does not address the metabolism of PET-tracers, and according to reference 31 “fluorine-18 present at the terminus of a straight saturated alkyl chain of two or more carbons appears quite resistant to defluorination in human subjects”. Additionally, reference 30 notes that deuteration – a standard approach in PET tracer development – can significantly reduce the defluorination susceptibility of [^{18}F]fluoroalkyl-based tracers. Finally, it is not clear why α,α -difluoroalkanes should generally be more stable than the corresponding fluoroalkanes. The authors should either provide more robust (preferably experimental) evidence to substantiate this point or consider removing it from the manuscript.

We agree that we don't presently have substantial experimental evidence for the metabolic stability (i.e. resistance to radiodefluorination) for the ^{18}F -labeled α,α -difluoroalkyl moiety. To make the discussion more relevant, and avoid undue speculation, we removed the following entire paragraph (as suggested by the Reviewer).

“Installation of NCA fluorine-18 at aliphatic carbons in PET tracers has generally been confined to the syntheses of [^{18}F]alkyl fluorides through the nucleophilic substitution of a good leaving group (3,28). Notable PET tracers that have been produced in this manner include [^{18}F]FDG, [^{18}F]AV-45, and [^{18}F]fallypride (8). However, in many cases candidate tracers carrying ^{18}F -fluoroalkyl groups undergo considerable radiodefluorination *in vivo* leading to an accumulation of the resulting [^{18}F]fluoride ion in skull and in other areas of the skeleton (10). This accumulation may hamper accurate quantification of tracer uptake in tissue near bones, especially for PET

imaging targets within brain (10,29-31). In this context, an ^{18}F - α,α -difluoromethyl moiety becomes an attractive feature for inclusion in a PET tracer design because of its expected greater resistance to defluorination *in vivo*. For example, the herbicide sulfentrazone metabolizes extensively but without defluorination of its α,α -difluoromethyl moiety (32). In this regard, the difluoromethyl group has higher C–F bond dissociation energy than the fluoromethyl group (33–35).”

The reference section is now updated with deletion of references 32–35. We ensured that the revised citations of references are in the correct order.

2.1.2 Molar activity and chemical purity of the radiolabeled compounds

In the chromatograms provided in the Supporting Information, non-radioactive impurities – possibly the corresponding fluorovinyl derivatives – often appear very close to the product peaks (Rt difference < 0.5 min; difference between UV- and radioactivity signals approximately 0.1 min; see, e.g., ^{18}F -labeled 1, 13–15, 17, 20, 26-28, 33, 35, 36, and 41–43). This proximity should make it extremely difficult to separate the radiolabeled compounds from the side products, potentially affecting both the chemical purity and the apparent molar activity of the isolated products. Despite this, the quality control chromatograms (spiked with non-radioactive reference compounds) consistently display a single UV peak corresponding to the reference compound. To achieve this, did the authors fractionate the product peaks to isolate the desired compounds from side products? For greater clarity, the authors should provide HPLC traces of crude [^{18}F]1, [^{18}F]7, [^{18}F]13 and [^{18}F]14 spiked with the corresponding reference compounds to illustrate the feasibility of separating the products from non-radioactive impurities. If complete separation is difficult to achieve, a discussion of this limitation and its potential impact on purity and molar activity should be included in the manuscript.

We thank Reviewer 3 for these comments on radiolabeled product purification and measurement of molar activity. A single unoptimized HPLC analysis method (HPLC condition A) was used for the rapid analyses of the crude radiofluorination reaction mixtures throughout our investigation of substrate scope. Under this HPLC condition, as pointed out by the Reviewer, in some cases the non-radioactive impurities elute closely with the desired [^{18}F] α,α -difluoromethylalkane product. Judicious optimization of HPLC conditions, such as the choices of stationary column, mobile phase composition, and gradient and flow rate, would be required to achieve baseline separation of the desired [^{18}F] α,α -difluoromethyl alkanes from the non-radioactive impurities. The efficiency of this optimization process may vary depending on the structural complexity of the substrate under investigation. By appropriate HPLC method

optimization, we expect that the vast majority of labeled products, if not all, can be completely separated from the non-radioactive vinyl fluoride impurities.

We did not fractionate product peaks to obtain [¹⁸F]**10**, [¹⁸F]**19**, [¹⁸F]**25**, [¹⁸F]**36**, and [¹⁸F]**39** in high purity, for molar activity measurements. We developed optimal ‘semi-preparative’ HPLC conditions for these substrates (“**HPLC conditions C**”) under section 3.1 of General HPLC methods in Supplementary Information and **Supplementary Figures 14, 18, 22, 26, and 30**). Under the optimized ‘semi-preparative conditions’, we were able to isolate the desired radioactive products completely free of the non-radioactive impurities and to satisfactorily measure their molar activities.

To address the recommendation of Reviewer 3 on example separations, we developed **HPLC condition D** for the analyses of the crude reaction mixtures of [¹⁸F]**1**, [¹⁸F]**7**, [¹⁸F]**13**, and [¹⁸F]**14**. By using a methanol-aqueous ammonium formate (100 mM) mobile phase and a slower gradient, we were able to achieve baseline separation of [¹⁸F]**1**, [¹⁸F]**7**, [¹⁸F]**13** from the pair of the non-radioactive *E/Z* vinyl fluorides, as shown in the newly added HPLC chromatograms (**Supplementary Figures 40, 46, and 59**). [¹⁸F]**14** was only partially separated from one of the two non-radioactive impurities (**Supplementary Figure 63**). The analytical HPLC chromatograms of non-radioactive **1, 7, 13, and 14** are placed underneath the analytical HPLC chromatograms of the crude reaction mixtures of [¹⁸F]**1**, [¹⁸F]**7**, [¹⁸F]**13** and [¹⁸F]**14** (same scale) to clearly illustrate that with optimal HPLC method development, complete separation of the desired radiolabeled compounds from the non-radioactive UV absorbing impurities is feasible (**Supplementary Figures 41, 47, 60, and 64**).

The additional experimental details and chromatographic data for the separation of [¹⁸F]**1**, [¹⁸F]**7**, [¹⁸F]**13**, and [¹⁸F]**14** are listed in Section 3.1.4. of **Supplementary Information** and in **Supplementary Figures 40 and 41, 46 and 47, 59 and 60 and 63/64**. The insets show expansion of the chromatograms near the desired radiolabeled products peaks to show the clear separation from non-radioactive impurities.

3.1.4. HPLC condition D

Column: Luna C18(2) (10 μm, 250 × 4.6 mm, 100 Å; Phenomenex).

The absorbance wavelength used for HPLC measurements was 254 nm.

Radioactivity was monitored using an in-line Flow-Count PMT radioactivity detector (Eckert & Ziegler).

Solvent A: aq. ammonium formate (100 mM); Solvent B: MeOH

Flow rate: 2 mL/min

0–5 min: 10 to 30% B

5–60 min: 30 to 80% B

Separation of [^{18}F]**1** using HPLC condition D.

Supplementary Figure 40. Analytical HPLC (condition D) chromatogram for unpurified [^{18}F]**1** with expanded section for 40–50 min shown in the inset.

Supplementary Figure 41. Analytical HPLC (condition D) chromatogram for reference **1**.

Separation of [^{18}F]7 using HPLC condition D.

Supplementary Figure 46. Analytical HPLC (condition D) chromatogram for unpurified [^{18}F]7 with expanded section for 40–50 min shown in the inset.

Supplementary Figure 47. Analytical HPLC (condition D) chromatogram for reference compound 7.

Separation of [^{18}F]13 using HPLC condition D.

Supplementary Figure 59. Analytical HPLC (condition D) chromatogram for unpurified [^{18}F]13 with expanded section for 30–40 min shown in the inset.

Supplementary Figure 60. Analytical HPLC (condition D) chromatogram for reference compound 13.

Separation of [^{18}F]14 using HPLC condition D.

Supplementary Figure 63. Analytical HPLC (condition D) chromatogram for unpurified [^{18}F]14 with expanded section for 30–35 min shown in the inset.

Supplementary Figure 64. Analytical HPLC (condition D) chromatogram for reference compound 14.

With these new results, we modified the Results and Discussion section on page 11 to include a brief discussion on purification approaches for the [¹⁸F]α,α-difluoromethylalkanes and added a cautionary note regarding potential impacts on purity and molar activity in case of partial separations. as follows:

“With respect to measuring molar activity, it is necessary to first remove all non-radioactive byproducts from the radioactive analyte, and especially non-radioactive vinyl products that may elute close to the desired radiolabeled product. Often, the labeled product is readily separable from non-radioactive byproducts by ‘semi-preparative’ reversed-phase HPLC after some method optimization regarding column type and elution conditions. We demonstrated the feasibility of complete separation of labeled products by single-pass ‘semi-preparative’ HPLC for [¹⁸F]10, [¹⁸F]19, [¹⁸F]25, [¹⁸F]36, and [¹⁸F]39 (Supplementary Figures 14, 18, 22, 26, and 30, respectively). HPLC analytical methods also need to be capable of separating carrier from likely chemical impurities, such as closely eluting elimination products. This was readily shown to be feasible with 3 of 4 tested examples, [¹⁸F]1, [¹⁸F]7, and [¹⁸F]13 (Supplementary Figures 40, 46, and 59).”

2.2 Minor points

2.2.1. Nomenclature consistency

The manuscript would benefit from a thorough revision to ensure adherence to the consensus nomenclature rules for radiopharmaceutical chemistry, as outlined in Coenen et al., Nuclear Medicine and Biology, 2017 [doi: 10.1016/j.nucmedbio.2017.09.004]. In Figures 1-5, “¹⁸F-” should be corrected to “[¹⁸F]F-” (or “[¹⁸F]fluoride”) to align with standard conventions.

We thank Reviewer 3 for advising the adherence of the standards on nomenclature. We have now thoroughly inspected the nomenclature throughout the manuscript and Supplemental Information for compliance and corrected where previously absent.

We corrected “¹⁸F- to [¹⁸F]F-” in Figures 1–5, and in all Supplementary Figures and Tables.

2.2.2. Data consistency and error reporting

The statement in the caption of Figure 5B, “Yields are mean ± range for n = 2, except for [¹⁸F]10 (mean ± SD for n = 3)” is misleading and appears to be incorrect. According to the Supporting Information (SI; p. S52-53), the data for [¹⁸F]10 also seem to be calculated from only two values, rather than three.

We thank Reviewer 3 for pointing out this reporting error. Figure 5 legend is now revised to **“Automated synthesis of ¹⁸F-labeled α,α -difluoromethylalkanes and the determination of their molar activities.** (A). Automated radiosynthesis. AY = activity yield; n.d.c. = Non-decay corrected; EOS = End of radiosynthesis; ERP = End of radionuclide production. **Yields are mean \pm SD for $n = 2$.** (B). Control experiments for the synthesis of [¹⁸F]10. NR = No reaction.”

Additionally, the reported error values diverge from values calculated based on the individual data points provided in the SI. For instance, for [18F]19, the manuscript reports a yield (mean \pm range) of 0.85 ± 0.15 GBq, while my calculations based on the individual values in the SI (0.74 GBq and 0.96 GBq) give a range (maximum - minimum) of 0.22 GBq and a standard deviation (SD) of 0.16 GBq. Similarly, for [18F]25, the reported yield is 1.04 ± 0.33 GBq, but calculations based on the SI values (1.26 GBq and 0.81 GBq) result in a range of 0.45 GBq and an SD of 0.32 GBq. In these cases, the error values reported in the manuscript align more closely with the standard deviation than with the range, though the precise basis for these values remains unclear. Additionally, for [18F]10, [18F]36 and [18F]39, there are slight discrepancies between the reported means (1.15, 1.26 and 1.18) and means calculated based on the SI data (1.13, 1.24 and 1.19), potentially due to rounding differences. To improve clarity, I recommend revising the figure caption (or addressing this issue elsewhere in the manuscript) to specify whether the error values represent standard deviation, range, or another measure. Additionally, verifying that all reported mean values align with the SI data and clearly stating any rounding methods applied would ensure consistency and transparency in data reporting.

All the yields are now reported as mean \pm SD. The corresponding figure captions and data are revised accordingly.

2.2.3. Figs. 4 and 5 and text corrections

In the reaction scheme of Fig. 4, please remove the term “New candidate tracer”. Additionally, in Figs. 4 and 5, as well as in the corresponding sections of the manuscript, replace the term “analogs” with “derivatives”.

We now removed “New candidate tracer” from Figure 4. We replaced “analogs” with the term “derivatives” in Figures 4, 5 and elsewhere in the manuscript.”

2.2.4 Determination of molar activity

Please provide additional details in the Supporting Information regarding the determination of molar activity. Additionally, include data points for the experimental carrier amount on the calibration curves to enhance clarity and reproducibility.

We thank reviewer for the recommendations on determination of molar activity. We provided additional details in the Supplementary Information for the calculation of molar activity under the “Section 3.3. Basic principle of molar activity determination”. The revision includes improved experimental details and description of how calibration curves were constructed and the equations of molar activity calculation. We also included additional Supplementary Tables to provide associated carrier amounts calculated from the calibration curves of the corresponding [¹⁸F]**10**, [¹⁸F]**19**, [¹⁸F]**25**, [¹⁸F]**36**, and [¹⁸F]**39** productions. We updated supplementary tables for clarity. Calibration curves of **19**, **36**, and **39** were updated with additional data points to ensure the carrier amount values from the corresponding ¹⁸F-productions fall within the range of calibration curve range. The carrier amounts (red solid circles) for [¹⁸F]**10**, [¹⁸F]**19**, [¹⁸F]**25**, [¹⁸F]**36** and [¹⁸F]**39** from the automated syntheses are now co-plotted on the calibration curves. These additional details should greatly benefit readers to understand the experiments and be able to reproduce.

The numbering of **Supplementary Figures 7, 11, 15, 19, and 23** from original submission is now changed to “**Supplementary Figures 13, 17, 21, 25, and 29**, respectively, for the calibration curves of compounds **10, 19, 25, 36, and 39**.”

Supplementary Figures 17, 19 and 23 were updated with additional data points for the construction of calibration curve. We also slightly adjusted the molar activity in Supplementary Information and manuscript based on the last calibration curves.

As suggested by Reviewer 3, we now included “data points for the experimental carrier amount on the calibration curves to enhance clarity and reproducibility. The data points (**red solid circles**) in **Supplementary Figures 12, 13, 17, 21, 25 and 29** represent the experimentally calculated carrier amounts for the radiosynthesis of [¹⁸F]**10**, [¹⁸F]**19**, [¹⁸F]**25**, [¹⁸F]**36**, and [¹⁸F]**39**.”

Supplementary Tables 3, 4, 5, 6, and 7, each represent summary of radiosynthesis of [¹⁸F]**10**, [¹⁸F]**19**, [¹⁸F]**25**, [¹⁸F]**36**, and [¹⁸F]**39**, respectively. Each Table is now split into two Tables with modified information for further clarity. They are currently numbered as “**Supplementary Tables 5 and 6** ([¹⁸F]**10**), **7 and 8** ([¹⁸F]**19**), **9 and 10** ([¹⁸F]**25**), **11 and 12** ([¹⁸F]**36**), and **13 and 14** ([¹⁸F]**39**).”

The updated changes were shown under Section 3.3.

3.3. Molar activity determination

3.3.1. Basic principle of molar activity determination

To determine molar activity, we first established a calibration curve using a series of known amounts of the reference compound. On this curve, the X-axis represents the amount of the reference compound (X), with values between 20 ng to 500 ng. The Y-axis denotes the UV absorbance peak area at 254 nm (Y; in arbitrary units \times min), measured by HPLC. A linear regression fit provided a calibration equation ($Y=KX$), establishing a direct relationship between absorbance peak area and compound quantity. In the experimental analysis, we calculated the molar activity of the ^{18}F -labeled product by applying its radioactivity (GBq), molecular weight (M) and the UV peak area measured via HPLC to the following formula.

$$\text{Carrier (X)} = \frac{\text{UV peak area (Y)}}{K} \text{ (ng)}$$

$$A_m = \frac{\text{Activity}}{\text{Carrier+radioactive compound}} \approx \frac{\text{Activity}}{\text{Carrier}} = \frac{\text{Activity} \times K \times M \times 1000}{\text{UV peak area (Y)}} \text{ (GBq/mmol)}$$

3.3.2. Optimization for molar activity measurement

The radiofluorination followed general procedure 9. ‘Semi-preparative’ purification was carried out using HPLC condition C. Analytical HPLC for $[\text{}^{18}\text{F}]\mathbf{10}$ was carried out using HPLC condition B.

quantity (ng)	20	40	60	80	100
UV peak area	55859	113647	170316	226798	284771

Supplementary Figure 12. Calibration curve for authentic reference **10** for determination of the molar activity of [^{18}F]**10**. Data points for the calibration curve are in blue. Carrier amounts in [^{18}F]**10** analytes from Supplemental Table 4 are shown in red. Analytical HPLC condition B was used.

Supplementary Table 3. Optimization for radiosynthesis of [^{18}F]**10** under different conditions. AY = activity yield. n.d.c. = non-decay-corrected.

Entry	Precursor (μmol)	Et_4NHCO_3 (mg)	AY (n.d.c.) (GBq)
1	2.5	0.5	1.15
2	2.5	0.25	1.22
3	1.5	0.25	1.22

Supplementary Table 4. Molar activity (A_m) of [^{18}F]**10** under different conditions. EOS = end of radiosynthesis.

Entry	Activity (GBq)	UV peak area	Amount of carrier (ng)	M	A_m (EOS) (GBq/ μmol)
1	0.01343	268551	94.53	232.27	33
2	0.01878	241448	84.99	232.27	51
3	0.01499	132808	46.75	232.27	74

3.3.3. Automated radiosyntheses

3.3.3.1. Automated radiosynthesis of [¹⁸F]10

The automated radiosynthesis followed general procedure 9 with 4-(3-bromo-3-fluoropropyl)-1,1'-biphenyl (1.5 μmol, 0.44 mg) and Et₄NHCO₃ (0.25 mg). The semi-preparative HPLC purification was carried out using HPLC condition C. Analytical HPLC for [¹⁸F]10 was carried out using condition B.

Supplementary Figure 13. Calibration curve for authentic reference 10 to determine molar activity with data for molar activity determinations added. Data points for the calibration curve are in blue. Experimental carrier amount data points from Supplementary Table 6 for two molar activity measurements are in red. Analytical HPLC condition B was used.

Supplementary Table 5. Radiosynthesis of [¹⁸F]10. AY = activity yield. n.d.c. = non-decay corrected.

Entry	Starting activity (GBq)	Radiosynthesis time (min)	Purity (%)	AY (n.d.c.) (GBq)
1	14.5	66	>99	1.22
2	17.8	62	>99	1.04
Average				1.13 ± 0.13

Supplementary Table 6. The molar activity determination of [¹⁸F]10. EOS = end of radiosynthesis. ERP = end of radionuclide production.

Entry	Activity (GBq)	UV peak area	Amount of carrier (ng)	M	A_m (EOS) (GBq/ μ mol)	A_m (ERP) (GBq/ μ mol)
1	0.01499	132808	46.75	232.27	74	127
2	0.01500	162840	57.32	232.27	61	106
Average					68 ± 9	117 ± 15

3.3.3.2. Automated radiosynthesis of [^{18}F]19

The automated radiosynthesis followed general procedure 9 with 5-bromo-5-fluoropentyl 4-iodobenzoate (1.5 μmol , 0.62 mg) and Et $_4$ NHCO $_3$ (0.25 mg). The semi-preparative HPLC purification was carried out using HPLC condition C. Analytic HPLC for [^{18}F]19 was carried out using HPLC condition B.

Supplementary Figure 17. Calibration curve for authentic reference 19 to determine molar activity with data for molar activity determinations added. Data points for the calibration curve are in blue. Experimental carrier amount data points from Supplementary Table 8 for molar activity measurements are shown in red. Analytical HPLC condition B was used.

Supplementary Table 7. Radiosynthesis of [¹⁸F]19. AY = activity yield. n.d.c. = non-decay-corrected.

Entry	Starting activity (GBq)	Radiosynthesis time (min)	Purity (%)	AY (n.d.c.) (GBq)
1	15.7	70	>99	0.74
2	15.7	67	>99	0.96
Average				0.85 ± 0.16

Supplementary Table 8. The molar activity determination of [¹⁸F]19. EOS = end of radiosynthesis. ERP = end of radionuclide production.

Entry	Activity (GBq)	UV peak area	Amount of carrier (ng)	M	A_m (EOS) (GBq/μmol)	A_m (ERP) (GBq/μmol)
1	0.02725	173479	128.18	354.14	75	134
2	0.03364	186605	137.88	354.14	86	136
Average					81 ± 8	135 ± 1

3.3.3.3. Automated radiosynthesis of [^{18}F]25

The automated radiosynthesis followed general procedure 9 with 1-((5-bromo-5-fluoropentyl)oxy)naphthalene (1.5 μmol , 0.47 mg) and Et $_4$ NHCO $_3$ (0.25 mg). The semi-preparative HPLC purification was carried out using HPLC condition C. Analytical HPLC for [^{18}F]25 was carried out using HPLC condition B.

Supplementary Figure 21. Calibration curve for authentic reference 25 with added data for molar activity determinations. Data points for the calibration curve are in blue. Experimental carrier amount data points from Supplementary Table 10 for molar activity measurement are in red.

Supplementary Table 9. Radiosynthesis of [¹⁸F]25. AY = activity yield. n.d.c. = non-decay-corrected.

Entry	Starting activity (GBq)	Radiosynthesis time (min)	Purity (%)	AY (n.d.c.) (GBq)
1	17.9	68	>99	1.26
2	15.7	68	>99	0.81
Average				1.04 ± 0.32

Supplementary Table 10. The molar activity determination of [¹⁸F]25. EOS = end of radiosynthesis. ERP = end of radionuclide production.

Entry	Activity (GBq)	UV peak area	Amount of carrier (ng)	M	A_m (EOS) (GBq/μmol)	A_m (ERP) (GBq/μmol)
1	0.03297	16424	111.70	250.29	74	123
2	0.02187	12269	83.44	250.29	66	110
Average					70 ± 6	117 ± 9

3.3.3.4. Automated radiosynthesis of [^{18}F]36

The automated radiosynthesis followed General procedure 9 with 5-bromo-5-fluoropentyl (2*S*)-2-(6-methoxynaphthalen-2-yl)propanoate (1.5 μmol , 0.60 mg) and Et₄NHCO₃ (0.25 mg). The semi-preparative HPLC purification was carried out using HPLC condition C. Analytical HPLC for [^{18}F]36 was carried out using HPLC condition B.

Quantity (ng)	20	40	60	80	100
UV peak area	6961	15390	23327	30662	39607
Quantity (ng)	120	140	160	180	200
UV peak area	47375	55296	63010	70801	79243

Supplementary Figure 25. Calibration curve for authentic reference 36 to determine molar activity with data for molar activity determinations added. Data points for the calibration curve are in blue. Experimental carrier amount data points from Supplementary Table 12 for molar activity measurement are in red.

Supplementary Table 11. Radiosynthesis of [¹⁸F]36. AY = activity yield. n.d.c. = non-decay corrected.

Entry	Starting activity (GBq)	Radiosynthesis time (min)	Purity (%)	AY (n.d.c.) (GBq)
1	18.8	71	>99	1.18
2	15.7	66	>99	1.30
Average				1.24 ± 0.08

Supplementary Table 12. The molar activity determination of [¹⁸F]36. EOS = end of radiosynthesis. ERP = end of radionuclide production.

Entry	Activity (GBq)	UV peak area	Amount of carrier (ng)	M	A_m (EOS) (GBq/μmol)	A_m (ERP) (GBq/μmol)
1	0.04993	59524	151.10	336.38	111	189
2	0.02537	41229	104.66	336.38	82	137
Average					97 ± 21	163 ± 37

3.3.3.5. Automated radiosynthesis of [^{18}F]39

The automated radiosynthesis followed general procedure 9 with 5-bromo-5-fluoropentyl 3-(4,5-diphenyloxazol-2-yl)propanoate (1.5 μmol , 0.69 mg) and Et $_4$ NHCO $_3$ (0.25 mg). The semi-preparative HPLC purification was carried out using HPLC condition C. Analytical HPLC for [^{18}F]39 was carried out using HPLC condition B.

Quantity (ng)	20	40	60	80	100
UV peak area	10090.5	20040	29996.5	40000	50441.5
Quantity (ng)	200	300	400	500	
UV peak area	101639	157859	209989	262947	

Supplementary Figure 29. Calibration curve for authentic reference 39 to determine molar activity with data for molar activity determinations added. Data points for the calibration curve are

in blue. Experimental carrier amount data points from Supplemental Table 14 for A_m measurement are in red.

Supplementary Table 13. Radiosynthesis of [^{18}F]39. AY = activity yield. n.d.c. = non-decay corrected.

Entry	Starting activity (GBq)	Radiosynthesis time (min)	Purity (%)	AY (n.d.c.) (GBq)
1	14.2	71	>99	1.07
2	14.7	68	>99	1.30
Average				1.19 ± 0.16

Supplementary Table 14. The molar activity determination of [^{18}F]39. EOS = end of radiosynthesis. ERP = end of radionuclide production.

Entry	Activity (GBq)	UV peak area	Amount of carrier (ng)	M	A_m (EOS) (GBq/ μmol)	A_m (ERP) (GBq/ μmol)
1	0.03924	217938	416.32	399.44	38	62
2	0.02105	122320	233.66	399.44	36	60
Average					37 ± 1	61 ± 1